# GUIDED ACTOR-CRITIC: OFF-POLICY PARTIALLY OBSERVABLE REINFORCEMENT LEARNING WITH PRIVILEGED INFORMATION

## ABSTRACT

Real-world decision-making systems often operate under partial observability due to limited sensing or noisy information, which poses significant challenges for reinforcement learning (RL). A common strategy to mitigate this issue is to leverage privileged information—available only during training—to guide the learning process. While existing approaches such as policy distillation and asymmetric actor-critic methods make use of such information, they frequently suffer from weak supervision or suboptimal knowledge transfer. In this work, we propose Guided Actor-Critic (GAC), a novel off-policy RL algorithm that unifies privileged policy and value learning under a guided policy iteration framework. GAC jointly trains a fully observable policy and a partially observable policy using constrained RL and supervised learning objectives, respectively. We theoretically establish convergence in the tabular case and empirically validate GAC on challenging benchmarks, including Brax, POPGym, and HumanoidBench, where it achieves superior sample efficiency and final performance.

## 1 INTRODUCTION

In many real-world domains—ranging from robotics (Tang et al., 2025) and autonomous driving (Zhu & Zhao, 2021) to finance (Fischer, 2018), and multi-agent systems (Zhang et al., 2021)—decision-making agents must operate under partial observability. Limited or noisy sensing, occlusions, and constrained instrumentation prevent direct access to the true environment state and complicate both perception and control. Robotics offers a clear illustration: physical platforms frequently lack rich sensing (e.g., dense tactile arrays or high-fidelity proprioception) or face substantial sensor noise due to hardware cost and environmental disturbance. Importantly, however, richer signals are often available during development or in simulation (e.g., full simulator state, contact forces, or privileged diagnostics), and these training-time signals can be exploited to accelerate learning even when they are not available at deployment.

We view such problems as Partially Observable Markov Decision Processes (POMDPs) (Kaelbling et al., 1998) augmented with training-only privileged information (Vapnik & Vashist, 2009; Lambrechts et al., 2023). In this formulation, an agent must learn a policy that acts on limited observations at execution time, while leveraging additional state or side-information during training to improve sample efficiency and robustness. This POMDP-with-privileged-information perspective is broadly applicable: robotics is a natural and important example, but the same setup arises in other sequential decision problems where richer development-time data exists (e.g., richer lab measurements in healthcare, extra simulatable state in simulated environments, or additional diagnostic signals in industrial control).

Two main approaches have been proposed to leverage privileged information (Cai et al., 2024): The first is privileged *policy* learning, also known as expert policy distillation (Czarnecki et al., 2019) or teacher-student learning, where a teacher policy is trained using privileged inputs and then distilled into a student policy operating under partial observability. However, if the teacher policy fails to account for the student's limited observations, the distillation may lead to suboptimal performance (Warrington et al., 2020; Cai et al., 2024). The second is privileged *value* learning, where a value function (or Q-function) trained with privileged information is used to guide the learning of a partially

observable policy, commonly known as asymmetric actor-critic (Pinto et al., 2018). While this approach provides indirect supervision via the RL objective, it lacks the strong guidance that direct policy supervision can offer.

Recently, Guided Policy Optimization (GPO) (Li & Xie, 2025) has been proposed as a more structured method that integrates both privileged *policy* and *value* learning. GPO jointly trains a privileged policy and a partially observable policy, enforcing alignment between the two. This setup offers more effective supervision while mitigating the shortcomings of distillation alone. However, GPO typically relies on on-policy samples to maintain behavioral consistency between the two policies, which significantly limits sample efficiency, especially in expensive robotic settings.

In this paper, we introduce a novel off-policy algorithm that exploits privileged information during training. We formulate a guided policy iteration framework that jointly optimizes a privileged policy via a constrained RL objective and a partially observable learner policy via a supervised learning objective. We show that, in the tabular setting, iterative evaluation and improvement lead both policies to converge to the same optimal solution. Building on this foundation, we propose a practical deep RL algorithm called **Guided Actor-Critic (GAC)**, which approximates this framework using neural networks. We validate our approach on several challenging benchmarks, including Brax (Freeman et al., 2021), POPGym (Morad et al., 2023), and HumanoidBench (Sferrazza et al., 2024). Our results demonstrate that GAC can effectively utilize privileged information in complex POMDPs with high sample efficiency.

## 2 BACKGROUND

### 2.1 NOTATION

We consider a Partially Observable Markov Decision Process (POMDP) (Kaelbling et al., 1998), defined by the tuple $\langle \mathcal{S}, \mathcal{A}, r, \mathcal{P}, \mathcal{O}, \gamma \rangle$, where $\mathcal{S}$ is the set of states, $\mathcal{A}$ is the set of actions, $r$ is the reward function, $\mathcal{P}$ is the transition probability function, $\mathcal{O}$ is the observation function, and $\gamma$ is the discount factor. At each time step $t$, the agent receives a partial observation $o_t \sim \mathcal{O}(\cdot \mid s_t)$ of the underlying state $s_t \in \mathcal{S}$. Based on $o_t$ or the full action-observation history $\tau_t = \{o_0, a_0, o_1, a_1, \ldots, o_t\}$, the agent selects an action $a_t \in \mathcal{A}$. The environment then transitions to the next state $s_{t+1} \sim \mathcal{P}(s_{t+1} \mid s_t, a_t)$, and the agent receives a reward $r_t = r(s_t, a_t)$. We denote by $\rho_\pi(s_t)$ and $\rho_\pi(s_t, a_t)$ the state and state-action marginals of the trajectory distribution induced by a policy $\pi(a_t \mid \tau_t)$. The agent's objective is to find an optimal policy $\pi^* : \tau \to \Delta(\mathcal{A})$ that maximizes the expected return:

$$J(\pi) = \sum_{t=0}^{T} \mathbb{E}_{(s_t, a_t) \sim \rho_\pi}[r(s_t, a_t)]. \tag{1}$$

For notational simplicity, we use $s$ to refer to the true state or any form of privileged information available during training, and $o$ to represent the available information at execution time, including partial observations and history.

### 2.2 LEARNING WITH PRIVILEGED INFORMATION

The general idea of using additional information available only during training traces back to early work on learning with privileged information (Vapnik & Vashist, 2009). Following the categorization in (Cai et al., 2024), empirical approaches to RL with privileged information can be broadly divided into two paradigms: privileged *policy* learning and privileged *value* learning.

In privileged policy learning (also known as expert distillation (Chen et al., 2020; Nguyen et al., 2022; Margolis et al., 2021) or teacher-student learning (Lee et al., 2020; Miki et al., 2022; Shenfeld et al., 2023a)), the key idea is to exploit the fact that learning in a fully observable MDP is generally easier and more well-understood. These methods first train a privileged expert policy $\mu$ that has access to the full state $s$, and then distill its behavior into a partially observable policy $\pi$. The distillation objective can be formalized as:

$$\min_{\pi \in \Pi} \mathbb{E}_{s \sim d_\beta}[D(\mu(\cdot|s), \pi(\cdot|o))], \tag{2}$$

where $\beta$ is a given behavior policy, and $D$ is a divergence measure (e.g., KL divergence). While this approach appears intuitive and promising—since it directly supervises the student using the

expert policy—recent studies (Cai et al., 2024) have shown that the resulting policy can still be strictly suboptimal, even given unlimited data. We present an illustrative example from the classical *TigerDoor* problem (Littman et al., 1995) in Appendix B.

In contrast, privileged value learning, also known as asymmetric actor-critic (Pinto et al., 2018; Andrychowicz et al., 2020; Baisero & Amato, 2021), leverages privileged information in the value function (e.g., the Q-function) during training, while keeping the policy conditioned only on partial observations. Variants of this approach (Andrychowicz et al., 2020; Pinto et al., 2018; Baisero et al., 2022; Zhang et al., 2020) include asymmetric versions of PPO (Schulman et al., 2017), DDPG (Lillicrap et al., 2019), DQN (Mnih et al., 2015), and SAC (Haarnoja et al., 2018) . This approach is also widely used in multi-agent reinforcement learning under the centralized training with decentralized execution (CTDE) paradigm (Oliehoek et al., 2008; Kraemer & Banerjee, 2016), where privileged information (e.g., the joint observations of all agents) is naturally available during training but not during execution. However, a key limitation of privileged value learning is that it only provides *indirect* supervision to the policy through the RL objective, which may be less sample-efficient compared to the direct supervised signals provided by expert policies in privileged policy learning.

Another line of work attempts to reconstruct latent representations of privileged information from partial observations. This is common in vision-based robotic tasks, for example, inferring robot proprioception from camera images. While effective in certain applications, these methods often lack generality across broader POMDP settings. A more comprehensive discussion of related work is deferred to Appendix A.

### 2.3 GUIDED POLICY OPTIMIZATION

A recently proposed paradigm, known as Guided Policy Optimization (GPO) (Li & Xie, 2025) integrates both privileged policy and privileged value learning into a unified framework. GPO builds upon the ideas from Guided Policy Search (GPS) (Levine & Koltun, 2013b; Zhang et al., 2016a; Montgomery & Levine, 2016), leveraging a privileged policy (referred to as the guider $\mu$) and a privileged value function to guide the training of a partially observable policy (referred to as the learner $\pi$).

Concretely, GPO co-trains the guider and learner jointly: the guider is trained using PPO under full state observability, and the learner is trained via supervision from the guider. Critically, GPO introduces a constraint on the divergence between the guider and learner policies. This ensures that the guider remains close enough to the learner's behavior to provide meaningful and effective guidance. This setup allows GPO to inherit the theoretical guarantees of privileged value learning, while also framing the supervision process from the guider as a form of privileged policy learning. In this sense, GPO can be viewed as a hybrid approach that unifies the strengths of both paradigms. However, since GPO is built upon trust-region methods like PPO, it is inherently an on-policy algorithm, which may lead to lower sample efficiency compared to off-policy alternatives.

## 3 METHOD

In this section, we introduce our off-policy guided actor-critic algorithm. Our approach builds upon the divergence-augmented policy iteration framework proposed by (Wang et al., 2019). We begin by presenting a theoretical derivation of our method, verify its convergence to the optimal policy within the policy class, and then describe a practical algorithm motivated by this theory. Following the convention in GPO, we refer to the privileged policy as the *guider* and the partially observable policy as the *learner*.

### 3.1 GUIDED POLICY ITERATION

Our method shares the core principle of GPO—co-training the guider and learner while keeping them closely aligned, so that the learner benefits from the supervision provided by the more informed guider. Therefore, we formalize the guider's objective as a constrained reinforcement learning problem:

$$J(\mu) = \sum_{t=0}^{T} \mathbb{E}_{(s_t, a_t) \sim \rho_\mu} \left[ r(s_t, a_t) \right] \ s.t. \ D_{\text{KL}}(\mu(\cdot|s_t) || \pi(\cdot|o_t)) \le \epsilon, \tag{3}$$

where the KL constraint ensures alignment between the guider and learner policies. Since solving the above constrained problem directly is difficult, we adopt a more tractable soft-constrained formulation:

$$J(\mu) = \sum_{t=0}^{T} \mathbb{E}_{(s_t, a_t) \sim \rho_\mu} \left[ r(s_t, a_t) - \alpha D_{\text{KL}}(\mu(\cdot|s_t)||\pi(\cdot|o_t)) \right], \tag{4}$$

where $\alpha$ is a tunable coefficient that controls the strength of the KL regularization. Based on this, we derive a general guided policy iteration algorithm that alternates between *policy evaluation* and *policy improvement* for both the guider and learner.

In the policy evaluation step of guided policy iteration, we need to estimate the value of any policy pair $(\mu, \pi)$ according to the objective in equation 5. For fixed $\mu$ and $\pi$, the *guided Q-value* can be computed iteratively via a modified Bellman backup operator:

$$\mathcal{T}^{\mu,\pi} Q^{\mu,\pi}(s_t, a_t) = r(s_t, a_t) + \gamma \mathbb{E}_{s_{t+1} \sim p}[V(s_{t+1})], \tag{5}$$

where the *guided state value* is defined as:

$$V(s_t) = \mathbb{E}_{a_t \sim \mu}[Q^{\mu,\pi}(s_t, a_t) - \alpha \log \frac{\mu(a_t|s_t)}{\pi(a_t|o_t)}]. \tag{6}$$

By repeatedly applying $\mathcal{T}^{\mu,\pi}$, we can obtain the converged guided Q-value function for the given policies:

**Lemma 3.1 (Guided Policy Evaluation).** *Let $\mathcal{T}^{\mu,\pi}$ be the Bellman backup operator, and let $Q^0 : \mathcal{S} \times \mathcal{A} \to \mathbb{R}$ be any initial function with $|\mathcal{A}| < \infty$. Define $Q^{k+1} = \mathcal{T}^{\mu,\pi} Q^k$. Then the sequence $Q^k$ converges.*

*Proof.* See Appendix C. $\qquad\square$

In the policy improvement step, we update both the guider and the learner using the estimated Q-values. The guider is updated by minimizing the KL divergence to the learner policy modulated by the exponential of the Q-function:

$$\mu_{\text{new}}(\cdot|s_t) = \arg\min_{\mu \in \Pi_\mu} D_{\text{KL}} \left( \mu(\cdot|s_t) \middle\| \frac{\pi_{\text{old}}(\cdot|o_t) \exp(\frac{1}{\alpha} Q^{\mu_{\text{old}}, \pi_{\text{old}}}(s_t, \cdot))}{Z(s_t)} \right), \tag{7}$$

where $\Pi_\mu$ is the policy class of $\mu$, and $Z(s_t)$ is the partition function which normalizes the distribution and can be ignored for gradient-based optimization.

The learner is then updated to minimize the KL divergence to the new guider policy:

$$\pi_{\text{new}}(\cdot|o_t) = \arg\min_{\pi \in \Pi_\pi} D_{\text{KL}} \big( \mu_{\text{new}}(\cdot|s_t) \big\| \pi(\cdot|o_t) \big), \tag{8}$$

where $\Pi_\pi$ is the learner's policy class. This two-step update leads to performance improvement respect to the objective in equation 4, as stated below:

**Lemma 3.2 (Guided Policy Improvement).** *Let $\mu_{old} \in \Pi_\mu$, $\pi_{old} \in \Pi_\pi$, and let $\mu_{new}, \pi_{new}$ be the solutions to equation 7 and equation 8. Then:*

$$Q^{\mu_{new}, \pi_{new}}(s_t, a_t) \geq Q^{\mu_{old}, \pi_{old}}(s_t, a_t) \quad \forall (s_t, a_t) \in \mathcal{S} \times \mathcal{A} \tag{9}$$

*Proof.* See Appendix C. $\qquad\square$

The full guided policy iteration algorithm alternates between the guided policy evaluation and guided policy improvement steps, and it will provably converge to the policy $\mu^*$ and $\pi^*$ in tabular case, as formally described in the following theorem:

**Theorem 3.3 (Guided Policy Iteration).** *Repeated application of guided policy evaluation (Lemma 3.1) and guided policy improvement (Lemma 3.2) from any $\mu \in \Pi_\mu$ and $\pi \in \Pi_\pi$ converges to policy $\mu^*$ and $\pi^*$.*

*Proof.* See Appendix C. $\qquad\square$

One key distinction between our Guided Policy Iteration and standard policy iteration is the presence of two simultaneously updated policies: the *guider* and the *learner*. To understand their respective convergence behaviors, recall that the guider policy $\mu$ takes the privileged state $s$ as input, whereas the learner policy $\pi$ relies only on the partial observation $o$. If both policies share the same parametrization, then it follows naturally that $\Pi_\pi \subseteq \Pi_\mu$, since the guider—having access to more informative inputs—possesses strictly greater representational capacity.

To obtain a closed-form characterization of convergence, we introduce a simplifying assumption: $\Pi_\mu$ is expressive enough to drive the KL divergence in equation 7 to zero. This assumption is generally reasonable, as it merely requires that the privileged guider be capable of imitating a Q-value–modulated version of the learner policy. Under this assumption, the update rule reduces to one involving only the learner policy $\pi$, as formalized in the following lemma:

**Lemma 3.4.** *Suppose $\Pi_\mu$ is expressive enough such that the KL divergence in equation 7 can be minimized to zero. Then, the policy improvement step for the learner policy $\pi$ can be reformulated as:*

$$J(\pi) = \mathbb{E}_{a_t \sim \pi_{old}}\big[\exp(\frac{1}{\alpha}Q^{\mu_{old},\pi_{old}}(s_t,a_t))\log \pi(a_t|o_t)\big]. \tag{10}$$

*Eventually, both the guider and learner policies converge to the same optimal policy:*

$$\pi^* = \mu^* = \arg\max_{\pi \in \Pi_\pi}\mathbb{E}_{a_t \sim \pi}[Q^*(s_t,a_t)], \tag{11}$$

*where $Q^*$ denotes the optimal Q-function.*

*Proof.* See Appendix C. □

This lemma indicates that the guider and learner will ultimately converge to the same optimal policy with respect to expected return. In contrast to maximum entropy reinforcement learning (Haarnoja et al., 2018)—where the optimal policy is biased when $\alpha \neq 0$—our guided policy iteration converges to the unbiased optimal policy, regardless of the choice of $\alpha$. Although the learning dynamics during training are affected by $\alpha$, the convergence guarantee makes the algorithm more robust to its tuning.

It is also worth noting that our framework shares conceptual similarities with the formulation of Maximum a Posteriori Policy Optimization (MPO) (Abdolmaleki et al., 2018), particularly when viewing its non-parametric auxiliary distribution as analogous to our guider policy. Further discussion is provided in Appendix D.

## 3.2 GUIDED ACTOR-CRITIC

The theoretical results discussed above are primarily applicable to tabular settings. To extend these ideas to large-scale continuous domains, we now introduce a practical algorithm based on function approximation and stochastic gradient optimization. We consider the following parameterized components: a guided Q-function $Q_\theta(s_t,a_t)$, a guider policy $\mu_\phi(a_t|s_t)$, and a learner policy $\pi_\psi(a_t|o_t)$. The parameters of these networks are denoted by $\theta$, $\phi$ and $\psi$, respectively.

The guided Q-function is trained by minimizing the Bellman residual:

$$J_Q(\theta) = \mathbb{E}_{(s_t,a_t,r_t,s_{t+1})\sim\mathcal{D},a_{t+1}\sim\mu}\left[\big(Q_\theta(s_t,a_t) - (r_t + \gamma(Q_{\bar\theta}(s_{t+1},a_{t+1}) - \alpha\log\frac{\mu_\phi(a_{t+1}|s_{t+1})}{\pi_\psi(a_{t+1}|o_{t+1})})))^2\right],$$
$$\tag{12}$$

where $\bar\theta$ denotes the parameters of a target Q-function, maintained as an exponential moving average of $\theta$ to stabilize training.

The policy parameters are optimized by minimizing the expected KL divergence as described in equation 13 and equation 14:

$$J_\mu(\phi) = \mathbb{E}_{s_t\sim\mathcal{D},a_t\sim\mu}\big[\alpha D_{\text{KL}}(\mu_\phi(\cdot|s_t)\|\pi_\psi(\cdot|o_t)) - Q_\theta(s_t,a_t)], \tag{13}$$

$$J_\pi(\psi) = \mathbb{E}_{s_t\sim\mathcal{D}}\big[D_{\text{KL}}(\mu_\phi(\cdot|s_t)\|\pi_\psi(\cdot|o_t))]. \tag{14}$$

In addition to learning from the guider via KL supervision, the learner can also benefit from reinforcement learning using trajectories collected by the guider policy, as the two policies are closely aligned.

To evaluate the learner's performance under this setting, we introduce an additional Q-function $Q_\varphi(s_t, a_t)$ trained by the following:

$$J'_Q(\varphi) = \mathbb{E}_{(s_t,a_t,r_t,s_{t+1})\sim\mathcal{D},a_{t+1}\sim\pi}\left[\left(Q_\varphi(s_t, a_t) - (r_t + \gamma Q_{\bar\varphi}(s_{t+1}, a_{t+1}))\right)^2\right], \quad (15)$$

and modify the learner's objective accordingly:

$$J_\pi(\psi) = \mathbb{E}_{s_t\sim\mathcal{D},a_t\sim\pi}\left[\alpha D_{\mathrm{KL}}(\mu_\phi(\cdot|s_t)\|\pi_\psi(\cdot|o_t)) - Q_\varphi(s_t, a_t)\right]. \quad (16)$$

The temperature parameter $\alpha$ controls the strength of KL regularization, but tuning it can be challenging since reward magnitudes may vary significantly over time and across tasks. Instead, we adopt an automatic adjustment mechanism that adapts $\alpha$ to match a target KL divergence $\epsilon$, which is easier to specify:

$$J(\alpha) = \mathbb{E}_{s_t\sim\mathcal{D}}[-\alpha D_{\mathrm{KL}}(\mu_\phi(\cdot|s_t)\|\pi_\psi(\cdot|o_t)) - \alpha\epsilon], \quad (17)$$

where $\epsilon$ is a predefined target KL divergence.

The complete algorithm is summarized in Algorithm 1, where we employ clipped double Q-learning to mitigate overestimation bias, following prior work (Haarnoja et al., 2018; Fujimoto et al., 2018). The method alternates between collecting experiences using the guider policy and updating the networks using mini-batches sampled from a replay buffer. Unlike GPO, the proposed method can leverage off-policy data while ensuring that the guider remains imitable. Further implementation details are provided in Appendix E.

## 4 EXPERIMENTS

In this section, we evaluate the empirical performance of GAC across various domains. Section 4.1 analyzes GAC on partially observable and noisy continuous control tasks in the Brax environment (Freeman et al., 2021). Section 4.2 examines GAC's performance on memory-based tasks from POPGym (Morad et al., 2023). Section 4.3 presents results on more challenging high-dimensional tasks from HumanoidBench (Sferrazza et al., 2024). Finally, Section 4.4 provides a discussion about the limitation of GAC. Additional ablation studies are provided in Appendix F.3.

### 4.1 CONTINUOUS CONTROL TASKS IN BRAX

We evaluate GAC and baselines on several classic continuous control tasks in Brax under partial observability and observation noise. We treat joint velocities as privileged information accessible only during training. To simulate sensor inaccuracies, we add Gaussian noise with zero mean and standard deviation $\sigma$ to the partial observations. We compare GAC against TGRL (Shenfeld et al., 2023b) (privileged policy learning), asymmetric SAC, GPO-clip (Li & Xie, 2025), RMA (Kumar et al., 2021) (representation learning), and standard SAC (Haarnoja et al., 2018). Figure 1 reports the performance of all methods across different noise levels over 2M environment steps. For fairness, we allocate 1M steps for teacher pretraining in TGRL and RMA and another 1M for student training. Since GPO-clip is on-policy, it is evaluated after 10M steps.

As shown in the figure, GAC consistently outperforms the baselines in both sample efficiency and final performance. Several observations emerge: First, privileged policy learning methods such as TGRL are less suitable for POMDPs with privileged information. This is due to their reliance on a pre-trained teacher, which often leads to suboptimal student policies since the teacher leverages privileged observations unavailable to the student. Notably, as the noise level increases, the same pre-trained teacher (each row) becomes progressively less effective—TGRL's performance degrades significantly (e.g., *Ant* tasks with $\sigma = 0.2, 0.3$), even underperforming SAC in more difficult settings (e.g., *HumanoidStandup*). This suggests that the cost of teacher pretraining may be unjustified in highly asymmetric settings. Second, RMA also performs poorly because privileged representations cannot be faithfully reconstructed from noisy partial observations. Once reconstruction fails, the policy receives inputs outside its training distribution and may behave arbitrarily. Third, SAC-asym demonstrates relatively stable performance, but its performance gap with GAC indicates the limitations of using privileged information solely in value estimation, a point we explore further in the next subsection. Last, although both GPO and GAC exploit privileged policy and value information, GAC benefits significantly from off-policy learning, achieving superior sample efficiency. For instance, GAC's

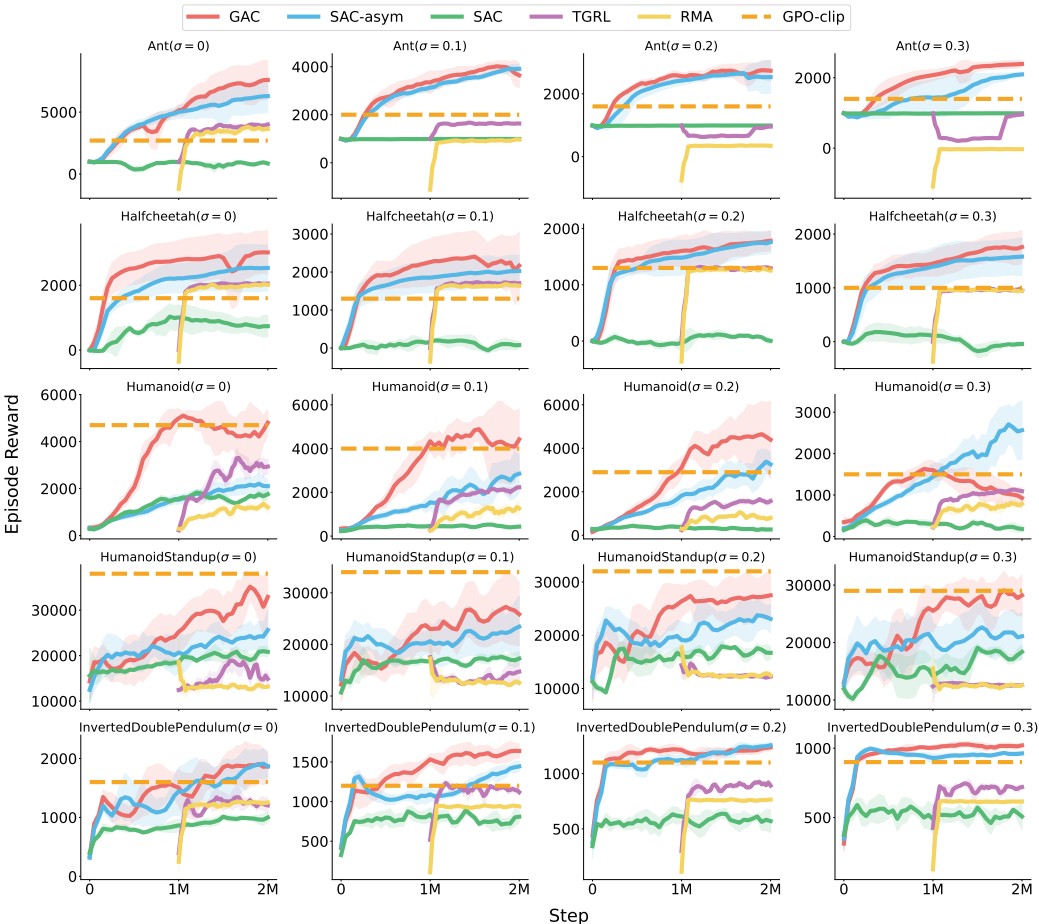

Figure 1: Performance comparison of GAC, SAC-asym, SAC, TGRL and GPO-clip on Brax. Partial observations are corrupted with Gaussian noise $N(0, \sigma)$.

performance at 2M steps clearly surpasses GPO-clip at 10M steps, except on *HumanoidStandup* task. It is also worth noting that GAC experiences a performance drop on the *Humanoid* task with $\sigma = 0.3$, which highlights a known limitation of GAC discussed in Section 4.4.

An additional set of image-based experiments is provided in Appendix F.2, where the true state serves as privileged information and only images are available as observations.

## 4.2 MEMORY TASKS IN POPGYM

In this subsection, we evaluate GAC on a suite of memory-intensive tasks from the POPGym benchmark. These experiments are designed to assess the ability of GAC to train effective memory-based models in both the actor and critic networks—an essential capability for POMDPs, where agents must leverage historical information for decision-making. The selected tasks include various card and board games that require extracting relevant patterns from observation histories. Privileged information in this setting is constructed as a summarized recorder of the observation history; further implementation details can be found in Appendix F.

Figure 2 reports performance across 15 POPGym tasks, comparing GAC to asymmetric SAC and standard SAC. As shown, GAC consistently demonstrates superior sample efficiency across most tasks, with the exception of a few particularly challenging ones where all methods struggle. GAC's advantage stems from its formulation, where the partially observable learner is directly supervised by the privileged guider, enabling more effective training in environments with long-term dependencies. In contrast, asymmetric SAC does not outperform standard SAC as significantly as it does in Brax, likely due to the limited utility of privileged value functions in memory-based settings. This highlights

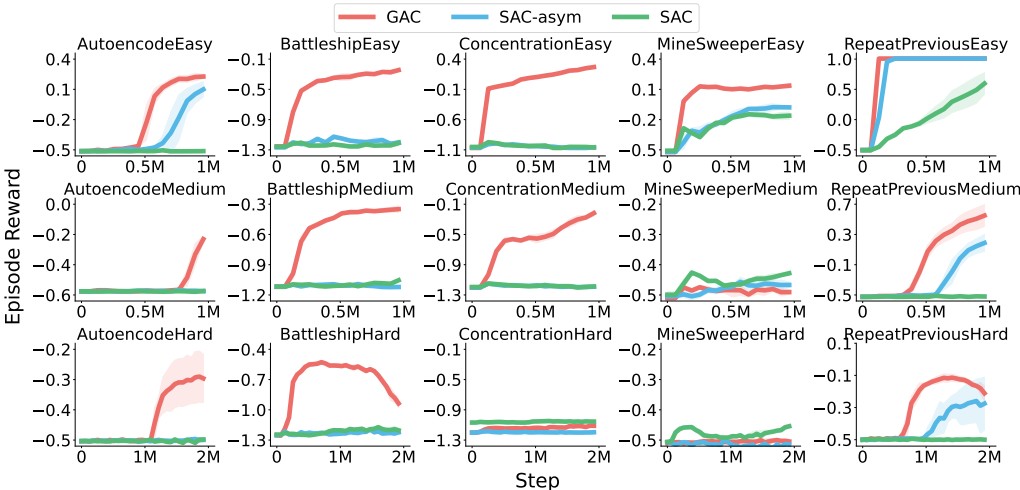

Figure 2: Performance comparison of GAC, SAC-asym, SAC on POPGym.

a key limitation of privileged value learning: since it only provides indirect supervision via the RL objective, it may be less effective than the direct guidance offered by a privileged policy. Additionally, GAC's success is partly attributed to the tight alignment between the guider and learner, avoiding the sub-optimality that can arise when the expert policy is too optimal (see Section 4.4).

## 4.3 CONTINUOUS CONTROL TASKS IN HUMANOIDBENCH

HumanoidBench is a high-dimensional simulated robotics benchmark featuring a humanoid robot equipped with dexterous hands, supporting a variety of challenging whole-body manipulation and locomotion tasks (Sferrazza et al., 2024).

We evaluate the algorithms on 8 manipulation tasks, where we retain all standard observations and additionally provide tactile feedback as privileged information during training, allowing us to evaluate how well algorithms can exploit such information. We also include 8 locomotion tasks. Since these are similar to the Brax setting, we report their results in Appendix F.2.

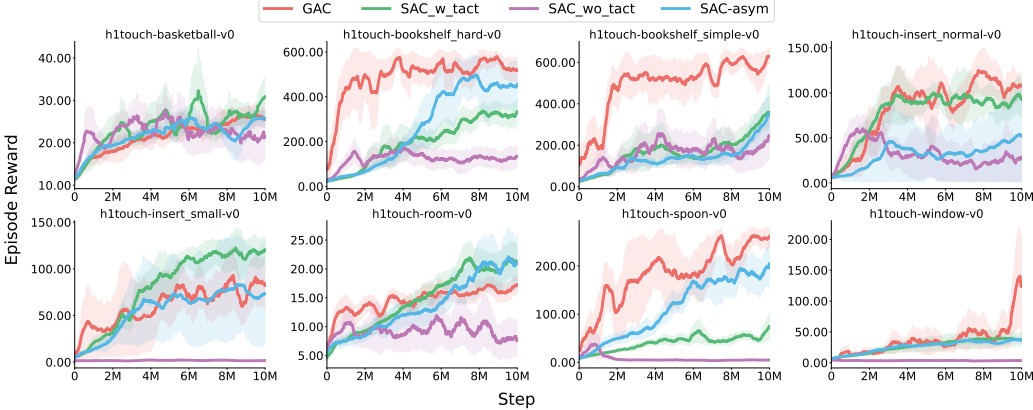

Figure 3: Performance comparison of GAC, SAC-asym, SAC_w_tact and SAC_wo_tact on manipulation tasks of HumanoidBench.

Figure 3 presents the performance of four algorithms on various manipulation tasks. Here, *SAC_w_tact* denotes SAC with tactile information available during both training and evaluation, while *SAC_wo_tact* refers to SAC trained and evaluated without any tactile input. These baselines serve to evaluate the contribution of tactile sensing to task performance. As illustrated in the figure, SAC_w_tact generally outperforms SAC_wo_tact across most tasks—except for *basketball*—underscoring the importance of tactile input for manipulation. Interestingly, in tasks such as *bookshelf_hard* and *spoon*, both GAC and asymmetric SAC—where tactile information is used only

during training—surpass the performance of SAC_w_tact. We hypothesize that this is due to the high dimensionality of the tactile data (over $10^3$), which far exceeds that of the standard observation space (typically around $10^2$). While tactile input is rich and informative, its complexity may hinder effective learning when used directly. Overall, GAC achieves substantial performance gains in scenarios where privileged information is available, demonstrating the potential of leveraging such information during training to enhance sample efficiency and policy effectiveness.

## 4.4 DISCUSSION

In this subsection, we discuss the limitations of GAC. In the *Humanoid* tasks with a high noise scale ($\sigma = 0.3$), we observe that GAC's performance unexpectedly deteriorates, despite strong results under lower noise levels. This degradation is illustrated in Figure 4, where the KL divergence between the guider and learner fails to converge to the desired value ($10^{-3}$). The underlying reason is that the regularization coefficient $\alpha$ must be bounded to prevent it from becoming excessively large and causing nu-

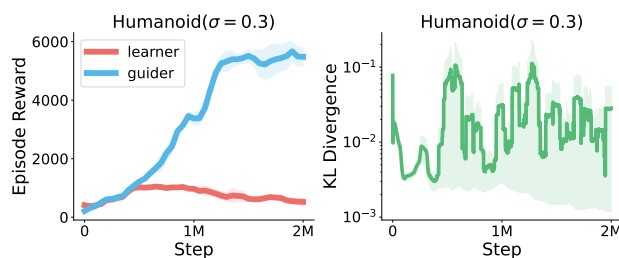

Figure 4: Performance comparison between the guider and learner in GAC (left). KL divergence between them (right).

merical instability. As a result, the ability to minimize the KL divergence is inherently limited. Nevertheless, we emphasize that such worst-case outcomes are rare. In our experiments, this issue appeared in only a single task, suggesting that while the limitation is genuine, it does not generally compromise the effectiveness of GAC. Addressing this challenge is left as an avenue for future work.

Another factor is the choice of the target KL divergence, which may influence GAC's performance. As shown in Figure 5, setting the target KL too high leads to an overly dominant guider that provides limited actionable feedback to the learner. Conversely, setting it too low results in an overly conservative guider that offers minimal advantage over the learner, thereby failing to guide effectively. Fortunately, GAC is generally robust as long as the target KL is selected

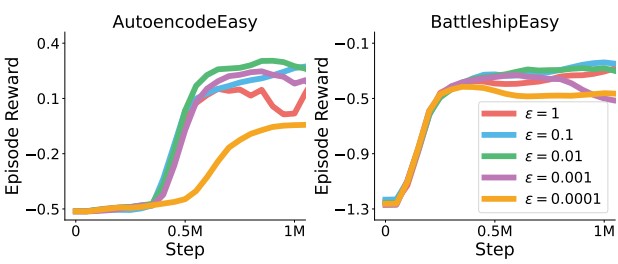

Figure 5: Performance of GAC with different target KL.

appropriately—for example, within the range $[0.001, 0.1]$. A practical heuristic is to tailor the target KL based on the degree of privileged information: the more privileged the information, the smaller the target KL should be. For instance, in the Brax domain, we adopt smaller target KL values for environments with higher noise levels (see Table 3). Similarly, in HumanoidBench, we use smaller target KLs for manipulation tasks (Table 6), where tactile sensing provides highly privileged observations.

## 5 CONCLUSION

We propose Guided Actor-Critic (GAC), an off-policy RL algorithm that leverages the strengths of both privileged policy learning and privileged value learning, while mitigating their respective limitations to achieve sample-efficient training. Our theoretical analysis introduces guided policy iteration, which we prove converges to the optimal policy. Based on this foundation, we derive the GAC algorithm and demonstrate empirically that it outperforms state-of-the-art methods in both privileged policy and value learning. These results highlight the potential of the guided RL framework for effectively exploiting privileged information in POMDPs.

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

## THE USE OF LARGE LANGUAGE MODELS (LLMS)

LLMs are used to polish the paper writing.

## A    RELATED WORKS

**Guided Policy Search (GPS)**. Guided Policy Search is a family of algorithms initially proposed by (Levine & Koltun, 2013b). Unlike direct policy search methods that optimize policy parameters end-to-end, GPS introduces an intermediate policy—typically a time-varying linear-Gaussian controller—learned via trajectory optimization. This controller then serves as a teacher for a parameterized neural network policy, which is trained through supervised learning. The GPS procedure consists of two key phases:

- **Control Phase**: A control policy interacts with the environment to minimize costs while ensuring learnability by the neural network policy.
- **Supervised Phase**: The neural network policy is trained via supervised learning to imitate the control policy.

In addition to the foundational works on GPS (Levine & Koltun, 2013a; Levine & Abbeel, 2014;?), our formulation is closely related to the approach in (Montgomery & Levine, 2016), which employs constrained LQR optimization for the control policies and supervised learning for the neural policy. This setup inherits the monotonic improvement guarantee of mirror descent (Beck & Teboulle, 2003), thereby ensuring consistent progress in policy performance. Over time, GPS has been extended in several directions, including integration with path integral methods (Chebotar et al., 2017b), combination with LQR techniques (Chebotar et al., 2017a), incorporation of memory models (Zhang et al., 2016a), and hybridization with model predictive control (Zhang et al., 2016b).

**Privileged Policy Learning**. Privileged policy learning, also known as expert distillation or teacher-student learning, refers to the paradigm where an expert policy—often with access to privileged information—is used to guide the learning of a student policy. A basic approach involves first training a privileged expert policy, followed by imitation learning techniques such as Behavioral Cloning (BC) (Pomerleau, 1991; Torabi et al., 2018) or DAgger (Ross et al., 2011). However, this two-stage method is often suboptimal, especially when the expert itself is suboptimal or when privileged information leads to behavior that is difficult to imitate directly. To address these limitations, recent approaches in policy distillation combine expert supervision with RL, jointly optimizing a composite objective that balances expert guidance and task reward (Schmitt et al., 2018; Czarnecki et al., 2018; 2019). For example, (Nguyen et al., 2022) integrate expert supervision into SAC (Haarnoja et al., 2018) by replacing the entropy term with a divergence between the student and expert policies. (Weihs et al., 2021) propose a dynamic mechanism that adjusts the balance between BC and RL based on the student's ability to imitate the expert. (Walsman et al., 2023) employ potential-based reward shaping (Ng et al., 1999) using the expert's value function to steer policy gradients. (Shenfeld et al., 2023b) augment the entropy term in SAC to blend expert guidance with task rewards, modulating the trade-off based on the student's relative performance.

While these methods can be applied to POMDPs with privileged information, most do not explicitly address the expert's training process, assuming instead that a high-quality expert is readily available. However, it has been shown that directly training a privileged expert without considering the student's limitations can lead to suboptimal outcomes for the student policy (Cai et al., 2024). Therefore, relying on such methods without carefully designing the expert or accounting for the cost of expert training may not yield the best results—particularly in settings where expert training is expensive or constrained.

**Privileged Value Learning**. Privileged value learning, also known as asymmetric actor-critic, leverages privileged information in the value function (or Q-function) during policy evaluation, while the policy itself operates under partial observations. This approach can be naturally extended from standard RL algorithms such as DQN (Mnih et al., 2015), PPO (Schulman et al., 2017), DDPG (Lillicrap et al., 2019), and SAC (Haarnoja et al., 2018). For instance, (Baisero et al., 2022) propose a model-based asymmetric policy iteration framework, later relaxed into a model-free variant based on DQN. (Andrychowicz et al., 2020) employ asymmetric PPO to learn dexterous in-hand manipulation policies that perform vision-based object reorientation using a physical Shadow Dexterous Hand. (Pinto et al., 2018) introduce asymmetric DDPG for image-based robotic control, where the critic has access to full state information while the actor learns from images alone. Similarly, (Killing et al., 2021) apply asymmetric SAC to address high-conflict scenarios in autonomous driving. In addition, privileged value learning is also widely adopted in cooperative multi-agent RL (Foerster et al., 2018; Lowe et al., 2017; Rashid et al., 2020; Yu et al., 2022) under the centralized training with decentralized execution (CTDE) paradigm (Oliehoek et al., 2008; Kraemer & Banerjee, 2016), where each agent must act based on its local observation and action history, while the critic can access global (privileged) information during training.

Compared to privileged policy learning, privileged value learning avoids issues related to the suboptimality and does not incur the cost of training a separate expert policy. However, since supervision is provided indirectly through the RL objective, it may be less sample-efficient than methods that leverage direct expert supervision.

**Privileged representation learning and world models.** This line of work attempts to reconstruct latent representations of privileged information from partial observations. This is common in vision-based robotic tasks, for example, Sermanet et al. (2018); Seo et al. (2023) use multi-view setups (e.g., image-based manipulation with additional camera views) to learn more informative embeddings. Others (Lee et al., 2020; Salter et al., 2021; Kumar et al., 2021; Qi et al., 2023) leverage privileged simulator states during training and design policies that operate on both observed and inferred states. Such methods typically require careful architectural design, domain knowledge, and feature engineering, or rely on favorable structural properties of the POMDP (e.g., decodability (Efroni et al., 2022)). While effective in certain applications, these methods often lack generality across broader POMDP settings.

**Privileged World Models.** Beyond model-free approaches, several works explore how privileged information can enhance model-based RL. Seo et al. (2023), for example, improve DreamerV2 (Hafner et al., 2020) by training a single-view policy representation using multi-view data. More recently, Informed Dreamer (Lambrechts et al., 2023) strengthens DreamerV3's (Hafner et al., 2024) representation learning and world modeling by predicting privileged information during training. Scaffolder (Hu et al., 2024) further leverages privileged sensing across multiple training-only components—including world models, critics, exploration policies, and representation learning—to improve the target policy in sensory scaffolding scenarios. TWIST (Yamada et al., 2023) introduces a teacher–student distillation framework in which a state-trained teacher world model supervises a vision-based student model trained with domain-randomized imagery, enabling efficient sim-to-real transfer in model-based RL.

**Co-training methods**. Co-training methods (Chang et al., 2015; Tangkaratt et al., 2021; Song et al., 2018; Yang et al., 2024) can be seen as an extension of privileged policy learning, where the teacher and student policies are trained simultaneously rather than separately. This joint training paradigm is particularly well-suited for POMDPs with privileged information, as it avoids the extra cost of pretraining the teacher and may potentially mitigate the suboptimality issues commonly associated with privileged policy learning. Co-training typically relies on shared experience or regularization between the teacher and student, enabling more synergistic learning. Several works have explored co-training in this context. For example, (Haklidir & Temeltaş, 2021) and (Salter et al., 2021) propose training two RL agents (using SAC and DDPG, respectively), where one has full observability and the other operates asymmetrically; the two agents alternate in collecting experiences and are optimized jointly. (Warrington et al., 2020) introduces adaptive asymmetric DAgger, where the expert is trained via RL and the student learns by imitation; a mixture of the two policies is used during data collection, following the DAgger framework. (Wu et al., 2025) co-trains a privileged teacher using PPO and a partially observable student through imitation, with both policies alternating their interaction with the environment.

However, many existing co-training approaches are empirical in nature and not explicitly designed to address the suboptimality induced by privileged policy learning. Recently, (Li & Xie, 2025) proposed Guided Policy Optimization (GPO), which combines the strengths of both privileged policy and value learning. GPO offers the same theoretical performance guarantees as privileged value learning while benefiting from the supervised structure of privileged policy learning. GPO follows a four-step iterative procedure:

- **Data Collection**: Collect trajectories by executing the guider's policy, denoted as $\mu^{(k)}$.
- **Guider Training**: Update the guider $\mu^{(k)}$ to $\hat{\mu}^{(k)}$ according to RL objective $V_{\mu^{(k)}}$.
- **Learner Training**: Update the learner to $\pi^{(k+1)}$ by minimizing the distance $D(\pi, \hat{\mu}^{(k)})$.
- **Guider Backtracking**: Set $\mu^{(k+1)}(\cdot|s) = \pi^{(k+1)}(\cdot|o)$ for all states $s$ before the next iteration.

The final step—guider backtracking—is the key distinction from prior co-training methods, ensuring the monotonic policy improvement property of GPO. Compared to our method GAC, both GPO and GAC can be interpreted within the broader framework of policy mirror descent (Beck & Teboulle, 2003; Tomar et al., 2020), where the guider acts as an intermediate step in the learner's policy update.

## B    TIGERDOOR EXAMPLE

Table 1: TigerDoor problem

| $s$ \ $a$ | $a_L$ | $a_R$ | $a_l$ |
|---|---|---|---|
| $s_L$ | 1 | 0 | $-0.1$ |
| $s_R$ | 0 | 1 | $-0.1$ |

The classical *TigerDoor* problem (Littman et al., 1995) describes a scenario in which a tiger is hidden behind one of two doors. The state space is $\mathcal{S} = \{s_L, s_R\}$, where $s_L$ and $s_R$ correspond to the tiger being behind the left or right door, respectively. The action space is $\mathcal{A} = \{a_L, a_R, a_l\}$, where $a_L$ and $a_R$ represent opening the left and right doors, and $a_l$ represents listening to determine the tiger's location. The payoff matrix is shown in Table 1.

Initially, the agent does not know the tiger's location unless it takes the listen action $a_l$. The optimal policy is to first choose $a_l$ (listen) and then open the door that has the tiger, yielding a reward of 0.9 in expectation. However, if the agent has access to privileged information—such as the exact location of the tiger—a teacher policy trained with this information will simply learn to open the correct door directly. This becomes problematic when using such a teacher to supervise a student policy that lacks access to the privileged information. The student may imitate the teacher by directly choosing a door without learning to listen, leading to a suboptimal policy. This example illustrates a key limitation of privileged policy learning: pretraining a teacher with access to privileged information can result in a policy that is not only unhelpful but potentially harmful when used to guide a student that operates under partial observability.

## C    PROOFS

**Lemma C.1** (**Guided Policy Evaluation**). *Consider the Bellman backup operator $\mathcal{T}^{\mu,\pi}$ and a mapping $Q^0 : \mathcal{S} \times \mathcal{A} \to \mathbb{R}$ with $|\mathcal{A}| < \infty$, and define $Q^{k+1} = \mathcal{T}^{\mu,\pi} Q^k$. Then the sequence $Q^k$ will converge.*

*Proof.* Define the KL divergence augmented reward as

$$r_{\pi,\mu}(s_t, a_t) = r(s_t, a_t) + \alpha D_{\text{KL}}(\mu(\cdot|s_t) || \pi(\cdot|o_t)) \tag{18}$$

and rewrite the update rule as

$$Q^{k+1}(s_t, a_t) = r_{\pi,\mu}(s_t, a_t) + \gamma \mathbb{E}_{s_{t+1} \sim p, a_{t+1} \sim \pi}[Q^k(s_{t+1}, a_{t+1})] \tag{19}$$

and apply the standard convergence results for policy evaluation. The assumption $D_{\mathrm{KL}}(\mu(\cdot|s_t)||\pi(\cdot|o_t)) < \infty$ is required to guarantee that the divergence augmented reward is bounded. □

**Lemma C.2** (**Guided Policy Improvement**). *Let $\pi_{old} \in \Pi_\pi$, $\mu_{old} \in \Pi_\mu$ and $\pi_{new}$, $\mu_{new}$ be the optimizer of the minimization problem defined by equation 7 and equation 8. Then $Q^{\mu_{new}, \pi_{new}}(s_t, a_t) \geq Q^{\mu_{old}, \pi_{old}}(s_t, a_t)$ for all $(s_t, a_t) \in \mathcal{S} \times \mathcal{A}$.*

*Proof.* Considering the definition of $\mu_{\mathrm{new}}$ in equation 7,

$$
\begin{aligned}
\mu_{\mathrm{new}}(\cdot|s_t) &= \arg\min_{\mu \in \Pi_\mu} \mathrm{D}_{\mathrm{KL}}\left(\mu(\cdot|s_t) \middle\| \frac{\pi_{\mathrm{old}}(\cdot|o_t) \exp(\frac{1}{\alpha} Q^{\mu_{\mathrm{old}}, \pi_{\mathrm{old}}}(s_t, \cdot))}{Z(s_t)}\right) \\
&= \arg\min_{\mu \in \Pi_\mu} J_{\mu_{\mathrm{old}}, \pi_{\mathrm{old}}}(\mu(\cdot|s_t))
\end{aligned}
\tag{20}
$$

It must be the case that $J_{\mu_{\mathrm{old}}, \pi_{\mathrm{old}}}(\mu_{\mathrm{new}}(\cdot|s_t)) \leq J_{\mu_{\mathrm{old}}, \pi_{\mathrm{old}}}(\mu_{\mathrm{old}}(\cdot|s_t))$. Hence

$$
\mathbb{E}_{a_t \sim \mu_{\mathrm{new}}}\left[\alpha \log \frac{\mu_{\mathrm{new}}(a_t|s_t)}{\pi_{\mathrm{old}}(a_t|o_t)} - Q^{\mu_{\mathrm{old}}, \pi_{\mathrm{old}}}(s_t, a_t)\right] \leq \mathbb{E}_{a_t \sim \mu_{\mathrm{old}}}\left[\alpha \log \frac{\mu_{\mathrm{old}}(a_t|s_t)}{\pi_{\mathrm{old}}(a_t|o_t)} - Q^{\mu_{\mathrm{old}}, \pi_{\mathrm{old}}}(s_t, a_t)\right],
\tag{21}
$$

where the partition function $Z(s_t)$ cancels.

Similarly, considering the definition of $\pi_{\mathrm{new}}$ in equation 7, we have

$$
\mathbb{E}_{a_t \sim \mu_{\mathrm{new}}}\left[\log \frac{\mu_{\mathrm{new}}(a_t|s_t)}{\pi_{\mathrm{new}}(a_t|o_t)}\right] \leq \mathbb{E}_{a_t \sim \mu_{\mathrm{new}}}\left[\log \frac{\mu_{\mathrm{new}}(a_t|s_t)}{\pi_{\mathrm{old}}(a_t|o_t)}\right].
\tag{22}
$$

As a result,

$$
\begin{aligned}
\mathbb{E}_{a_t \sim \mu_{\mathrm{new}}}\left[Q^{\mu_{\mathrm{old}}, \pi_{\mathrm{old}}}(s_t, a_t) - \alpha \log \frac{\mu_{\mathrm{new}}(a_t|s_t)}{\pi_{\mathrm{new}}(a_t|o_t)}\right] &\geq \mathbb{E}_{a_t \sim \mu_{\mathrm{new}}}\left[Q^{\mu_{\mathrm{old}}, \pi_{\mathrm{old}}}(s_t, a_t) - \alpha \log \frac{\mu_{\mathrm{new}}(a_t|s_t)}{\pi_{\mathrm{old}}(a_t|o_t)}\right] \\
&\geq V^{\mu_{\mathrm{old}}, \pi_{\mathrm{old}}}(s_t)
\end{aligned}
\tag{23}
$$

Next, consider the Bellman equation:

$$
\begin{aligned}
Q^{\mu_{\mathrm{old}}, \pi_{\mathrm{old}}}(s_t, a_t) &= r(s_t, a_t) + \gamma \mathbb{E}_{s_{t+1} \sim p}[V^{\mu_{\mathrm{old}}, \pi_{\mathrm{old}}}(s_t)] \\
&\leq r(s_t, a_t) + \gamma \mathbb{E}_{s_{t+1} \sim p}\left[\mathbb{E}_{a_t \sim \mu_{\mathrm{new}}}\left[Q^{\mu_{\mathrm{old}}, \pi_{\mathrm{old}}}(s_{t+1}, a_{t+1}) - \alpha \log \frac{\mu_{\mathrm{new}}(a_{t+1}|s_{t+1})}{\pi_{\mathrm{new}}(a_{t+1}|o_{t+1})}\right]\right] \\
&\quad \dots \\
&\leq Q^{\mu_{\mathrm{new}}, \pi_{\mathrm{new}}}(s_t, a_t)
\end{aligned}
\tag{24}
$$

□

**Theorem C.3** (**Guided Policy Iteration**). *Repeated application of guided policy evaluation (Lemma 3.1) and guided policy improvement (Lemma 3.2) from any $\mu \in \Pi_\mu$ and $\pi \in \Pi_\pi$ converges to policy $\mu^*$ and $\pi^*$.*

*Proof.* Let $\mu_i$ and $\pi_i$ be the policies at iteration $i$, By Lemma 3.2, the sequence $Q^{\mu_i, \pi_i}$ is monotonically increasing. Since $Q$ is bounded above for $\mu \in \Pi_\mu$ and $\pi \in \Pi_\pi$ (both the reward and entropy are bounded), the sequence converges to some $\mu^*$ and $\pi^*$. □

**Lemma C.4.** *Suppose $\Pi_\mu$ is expressive enough that the KL divergence in equation 7 can be minimized to zero. The policy improvement of learner policy $\pi$ can be viewed as:*

$$
J(\pi) = \mathbb{E}_{a_t \sim \pi_{old}}\left[\exp(\frac{1}{\alpha} Q^{\mu_{old}, \pi_{old}}(s_t, a_t)) \log \pi(a_t|o_t)\right],
\tag{25}
$$

*and finally the guider policy and learner policy will converge to the same optimal policy*

$$
\pi^* = \mu^* = \arg\max_{\pi \in \Pi} \mathbb{E}_{a_t \sim \pi}[Q^*(s_t, a_t)],
\tag{26}
$$

*where $Q^*$ is the optimal Q-function.*

*Proof.* By assumption, we have

$$\mu_{\text{new}}(\cdot|s_t) = \frac{\pi_{\text{old}}(\cdot|o_t)\exp(\frac{1}{\alpha}Q^{\mu_{\text{old}},\pi_{\text{old}}}(s_t,\cdot))}{Z(s_t)}. \tag{27}$$

Then, the update of the learner policy $\pi$ will be

$$
\begin{aligned}
\pi_{\text{new}} &= \arg\min_{\pi\in\Pi_\pi} D_{\text{KL}}\big(\mu_{\text{new}}(\cdot|s_t)\big\|\pi(\cdot|o_t)\big) \\
&= \arg\min_{\pi\in\Pi_\pi} \mathbb{E}_{a_t\sim\mu_{\text{new}}}[-\log\pi(a_t|o_t)] \\
&= \arg\max_{\pi\in\Pi_\pi} \mathbb{E}_{a_t\sim\pi_{\text{old}}}\big[\exp(\frac{1}{\alpha}Q^{\mu_{\text{old}},\pi_{\text{old}}}(s_t,a_t))\log\pi(a_t|o_t)\big],
\end{aligned}
\tag{28}
$$

where we drop the terms that unrelated to $\pi$.

Since the iteration converges to $\mu^*$ and $\pi^*$, by defining

$$J(\pi) = \mathbb{E}_{a_t\sim\pi}\big[\exp(\frac{1}{\alpha}Q^{\mu^*,\pi^*}(s_t,a_t))\log\pi(a_t|o_t)\big], \tag{29}$$

we know that $J(\pi^*) \geq J(\pi)$ for all $\pi \in \Pi$. Moreover, we can derive from equation 29 that $\pi^*$ is deterministic when $J(\pi)$ is maximized. Consequently, $\mu^*$ is deterministic and identical to $\pi^*$ based on equation 27.

Then, considering equation 20, we have

$$\mu^* = \pi^* = \arg\min_{\mu\in\Pi_\mu} J_{\mu^*,\pi^*}(\mu) = \arg\max_{\pi\in\Pi}\mathbb{E}_{a_t\sim\pi}[Q^*(s_t,a_t)]. \tag{30}$$

$\square$

## D  RELATIONSHIP TO MPO

Maximum a posterior Policy Optimization (MPO) (Abdolmaleki et al., 2018) is an actor-critic algorithm which employs KL-regularization in the policy optimization step. Specifically, the objective of MPO can be written as

$$J(q,\theta) = \mathbb{E}_q\bigg[\sum_{t=0}^{\infty}\gamma^t\big[r_t - \alpha D_{\text{KL}}\big(q(a|s_t)||\pi(a|s_t,\theta)\big)\big]\bigg] + \log p(\theta), \tag{31}$$

where $q$ is an auxiliary distribution and $p$ is a prior over policy parameters. If we drop the prior term and regard $q$ as the privileged guider $\mu$, the objective is the same for GAC. Moreover, the actual objective of MPO minimized by gradient descent takes the following form:

$$J(\theta) = \mathbb{E}_{s\sim\rho_{\theta'}}\bigg[\mathbb{E}_{a\sim\pi_{\theta'}}\bigg[\exp\bigg(\frac{1}{\eta}Q(s,a)\bigg)\log\pi_\theta(a|s)\bigg] - \alpha D_{\text{KL}}(\pi_{\theta'}(\cdot|s)||\pi_\theta(\cdot|s))\bigg], \tag{32}$$

which is also similar to the equation 10 in lemma 3.4. Although there are strong connections between the formulation GAC and MPO, there are several key differences. First, GAC deals with asymmetric observation, while MPO deals with regular MDPs. Second, MPO utilizes a non-parametric representation of $q$, while GAC's guider $\mu$ is explicitly parameterized. Third, MPO's behavioral policy is $\pi$ while GAC's is the guider $\mu$, which allows to potentially collect better trajectories. Last, in the policy evaluation step, MPO adopts standard off-policy evaluation, while GAC's is specialized for guided Q-value, which is similar to SAC.

## E  IMPLEMENTATION DETAILS

In this section, we present the implementation details of GAC. The pseudo code of our algorithm is provided in Algorithm 1, where we utilize six trainable networks, one for guider policy, one for learner policy, two Q-networks for guider and two Q-networks for learner. We utilize the guider to interact with environment and collect corresponding experience in the replay buffer, and execute update analogous to off-policy RL algorithm using the loss function defined in equation 12, equation 15, equation 13, equation 16 and equation 17.

Both policy networks parameterize Gaussian actions by outputting a mean and a standard deviation; actions are obtained by sampling from the Gaussian and applying a $tanh$ transform. Since the learner does not require active exploration at execution time, we share the standard-deviation parameter between guider and learner and stop gradients through the shared std when computing the KL-divergence; only the means are updated by the KL loss. For guider exploration we use the same entropy regularization scheme and target entropy as SAC. Additional low-level implementation details (network architectures, optimizer hyperparameters, seed handling) will be provided in the code release.

---

**Algorithm 1: Guided Actor-Critic**

---

**Input:** $\theta_1, \theta_2, \phi, \psi, \varphi_1, \varphi_2$ ;                                    // Initial parameters
$\bar{\theta}_1 \leftarrow \theta_1, \bar{\theta}_2 \leftarrow \theta_2, \bar{\varphi}_1 \leftarrow \varphi_1, \bar{\varphi}_2 \leftarrow \varphi_2$ ;          // Initialize target network
$\mathcal{D} \leftarrow \emptyset$ ;                                    // Initialize replay buffer
**for** *each iteration* **do**
   **for** *each environment step* **do**
      $\mathbf{a}_t \sim \mu_\phi(\mathbf{a}_t|s_t)$ ;                                    // Sample action from guider
      $s_{t+1} \sim p(s_{t+1}|s_t, \mathbf{a}_t)$ ;                                    // Sample transition
      $\mathcal{D} \leftarrow \mathcal{D} \cup \{(s_t, \mathbf{a}_t, r_t, s_{t+1})\}$ ;                                    // Store transition
   **end**
   **for** *each gradient step* **do**
      **for** $i \in \{1, 2\}$ **do**
         $\theta_i \leftarrow \theta_i - \lambda_Q \hat{\nabla}_{\theta_i} J_Q(\theta_i)$ ;                                    // Update Q-function through equation 12
         $\varphi_i \leftarrow \varphi_i - \lambda_Q \hat{\nabla}_{\varphi_i} J_Q(\varphi_i)$ ;                                    // Update Q-function through equation 15
      **end**
      $\phi \leftarrow \phi - \lambda_\mu \hat{\nabla}_\phi J_\mu(\phi)$ ;                                    // Update guider policy through equation 13
      $\psi \leftarrow \psi - \lambda_\pi \hat{\nabla}_\psi J_\pi(\psi)$ ;                                    // Update learner policy through equation 16
      $\alpha \leftarrow \alpha - \lambda_\alpha \hat{\nabla}_\alpha J(\alpha)$ ;    // Adjust temperature through equation 17
      **for** $i \in \{1, 2\}$ **do**
         $\bar{\theta}_i \leftarrow \tau\theta_i + (1-\tau)\bar{\theta}_i$ ;                                    // Update target network
         $\bar{\varphi}_i \leftarrow \tau\varphi_i + (1-\tau)\bar{\varphi}_i$ ;                                    // Update target network
      **end**
   **end**
**end**
**Output:** $\theta_1, \theta_2, \phi, \psi, \varphi_1, \varphi_2$ ;                                    // Optimized parameters

---

# F EXPERIMENTAL SETTINGS

## F.1 HYPERPARAMETERS

The hyperparameters used in the experiments from Section 4.1, 4.2, and 4.3 are listed in Table 2, Table 4, and Table 5, respectively. All SAC-based methods share the same set of core hyperparameters. For GAC, the only additional hyperparameter is the target KL divergence, which is selected from a predefined set. The specific target KL values for each task are detailed in Table 3 and Table 6. The heuristic for selecting the target KL is based on the asymmetry between the guider and learner observations When the privileged observation is substantially different from the partial observation (e.g., Brax tasks with high noise levels, or manipulation tasks in HumanoidBench), a smaller target KL is preferred. When the privileged observation can be partially inferred from the partial observation (e.g., tasks in POPGym), a larger target KL is appropriate.

Table 2: Hyperparameters of GAC and SAC in Brax.

| Parameter | Value |
|---|---|
| optimizer | Adam |
| learning_rate | 3e-4 |
| number_of_environments | 128 |
| number_of_timesteps | 2e6 |
| episode_length | 1000 |
| replay_buffer_size | 1e6 |
| discount ($\gamma$) | 0.99 |
| grad_update_per_step | 0.5 |
| target_smoothing_coefficient | 0.005 |
| maximum_gradient_norm | 1 |
| batch_size | 512 |
| actor_hidden_layers | [256, 256] |
| critic_hidden_layers | [256, 256] |
| activation | SiLU |
| target_entropy | $-0.5 * |\mathcal{A}|$ |
| target_kl | [0.01,0.005,0.001] |

Table 3: GAC Environment Specific Parameters in Brax.

| Environment | target_kl $[\sigma = 0, \sigma = 0.1, \sigma = 0.2, \sigma = 0.3]$ |
|---|---|
| Ant | $[0.01, 0.005, 0.005, 0.001]$ |
| HalfCheetah | $[0.01, 0.005, 0.001, 0.001]$ |
| Humanoid | $[0.005, 0.001, 0.001, 0.001]$ |
| HumanoidStandup | $[0.005, 0.005, 0.001, 0.001]$ |
| InvertedDoublePendulum | $[0.001, 0.001, 0.001, 0.001]$ |

## F.2 ADDITIONAL RESULTS

We provide the results of GAC, SAC-asym and DreamerV3 on 8 locomotion tasks in Figure 7. The partial observation is similar to Brax domain, where the velocity information of all joints is treated as privileged information and removed from the agent's observation. Frame stacking is not adopted since we tried and found no significant performance difference. We observe that GAC consistently outperforms asymmetric SAC, highlighting its superior ability to leverage privileged information in complex, high-dimensional continuous control settings. Although DreamerV3 sometimes achieves higher performance, it is a model-based approach and thus considerably more computationally demanding and slower than model-free methods.

We also report results of GAC and SAC-asym on six MuJoCo tasks in Figure 8. In this setting, the agent receives only image observations, while the privileged information corresponds to the robot's true state. This mirrors common real-world scenarios where only camera inputs are available to capture robot locomotion. Again, GAC outperforms asymmetric SAC, demonstrating the generality of our approach across different observation modalities.

## F.3 ADDITIONAL ABLATIONS

This subsection presents additional ablation studies for GAC. Recall that we use separate Q-functions for the guider and the learner, denoted as $Q_\theta$ and $Q_\varphi$, respectively. To examine the impact of this design choice, we evaluate a variant where the learner directly uses the guider's Q-function $Q_\theta$, referred to as GAC_share. We also assess the importance of the auxiliary RL loss for the learner by removing it—i.e., using Equation 14 instead of Equation 16—a variant we denote as GAC_wo_Q. As shown in Figure 6, GAC achieves better performance with both the auxiliary RL loss and separate Q-functions. While GAC_share performs comparably in most cases and outperforms GAC_wo_Q, the results suggest that the auxiliary RL loss significantly benefits learner training. Moreover, using a distinct Q-function for the learner provides a more accurate value estimation, since the guider's Q-function is biased from the learner's perspective.

Table 4: Hyperparameters of GAC and SAC in POPGym.

| Parameter | Value |
|---|---|
| optimizer | Adam |
| learning_rate | 3e-4 |
| number_of_environments | 1 |
| number_of_timesteps | 1e7 |
| replay_buffer_size | 1e6 |
| discount ($\gamma$) | 0.99 |
| grad_update_per_step | 1 / number_of_environments |
| batch_size | 32 * episode_length |
| actor_hidden_layers | [256, 256, 256(GRU), 256] |
| critic_hidden_layers | [256, 256, 256(GRU), 256] |
| activation | SiLU |
| target_entropy | $-0.9 * \log(1/|\mathcal{A}|)$ |
| target_kl | 0.05 |

Table 5: Hyperparameters of GAC and SAC in HumanoidBench.

| Parameter | Value |
|---|---|
| optimizer | Adam |
| learning_rate | 3e-4 |
| number_of_environments | 128 |
| number_of_timesteps | 2e6 |
| episode_length | 1000 |
| replay_buffer_size | 1e6 |
| discount ($\gamma$) | 0.99 |
| grad_update_per_step | 1 |
| target_smoothing_coefficient | 0.005 |
| maximum_gradient_norm | 1 |
| batch_size | 64 |
| actor_hidden_layers | [256, 256] |
| critic_hidden_layers | [256, 256] |
| activation | ReLU |
| target_entropy | $-0.5 * |\mathcal{A}|$ |
| target_kl (locomotion) | [0.02,0.01,0.005] |
| target_kl (manipulation) | [0.01,0.001,0.0001] |

## F.4 ENVIRONMENT DESCRIPTIONS

We provide a brief overview of the environments used and the pre-defined privileged information.

**Brax** (Freeman et al., 2021). Brax is an open source library for rigid body simulation with a focus on performance and parallelism on accelerators, written in JAX (Bradbury et al., 2018). The task in Brax contains a series of OpenAI gym-style MuJoCo-like tasks. We choose the *Ant*, *HalfCheetah*, *Humanoid*, *HumanoidStandup* and *InvertedDoublePendulum*. The privileged information is defined as the velocity and angular velocity of all joints, the dimension of the observation is described in Table 7

**POPGym** (Morad et al., 2023). POPGym contains a diverse collection of partially observable environments, where we choose some card games and broad games. We provide a brief description of the task and privileged information below:

- **Autoencode**: During the WATCH phase, a deck of cards is shuffled and played in sequence to the agent with the watch indicator set. The watch indicator is unset at the last card in the sequence, where the agent must then output the sequence of cards in order. The privileged information is the exact card that should be output at each timestep.

- **Battleship**: A partially observable version of Battleship game, where the agent has no access to the board and must derive its own internal representation. Observations contain

Table 6: GAC Environment Specific Parameters in HumanoidBench.

| Environment | target_kl |
|---|---|
| h1-basketball-v0 | 0.005 |
| h1-crawl-v0 | 0.02 |
| h1-highbar_simple-v0 | 0.005 |
| h1-hurdle-v0 | 0.01 |
| h1-maze-v0 | 0.01 |
| h1-pole-v0 | 0.01 |
| h1-slide-v0 | 0.02 |
| h1-walk-v0 | 0.01 |
| h1touch-basketball-v0 | 0.01 |
| h1touch-bookshelf_hard-v0 | 0.0001 |
| h1touch-bookshelf_simple-v0 | 0.0001 |
| h1touch-insert_normal-v0 | 0.0001 |
| h1touch-insert_small-v0 | 0.001 |
| h1touch-room-v0 | 0.01 |
| h1touch-spoon-v0 | 0.0001 |
| h1touch-window-v0 | 0.001 |

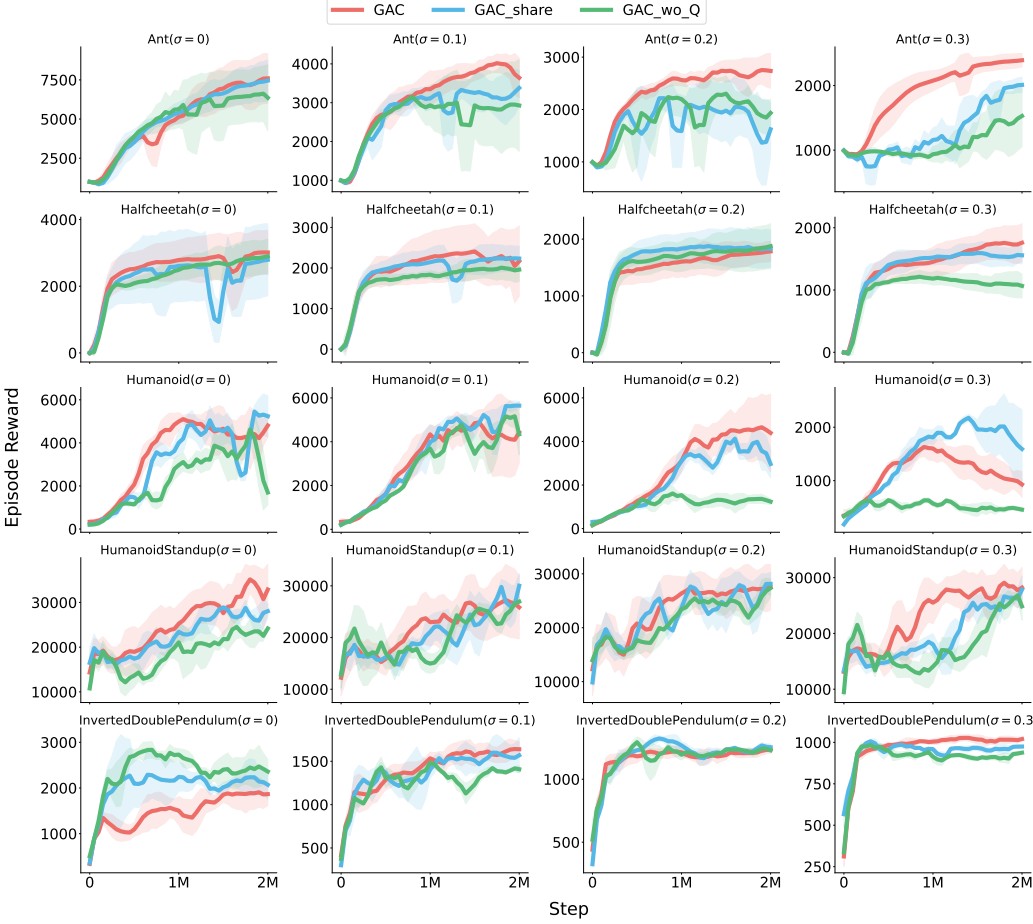

Figure 6: Ablation study of GAC on Brax.

either HIT or MISS and the position of the last salvo fired. The privileged information is a recorder that tracks all previous actions taken by the agent.

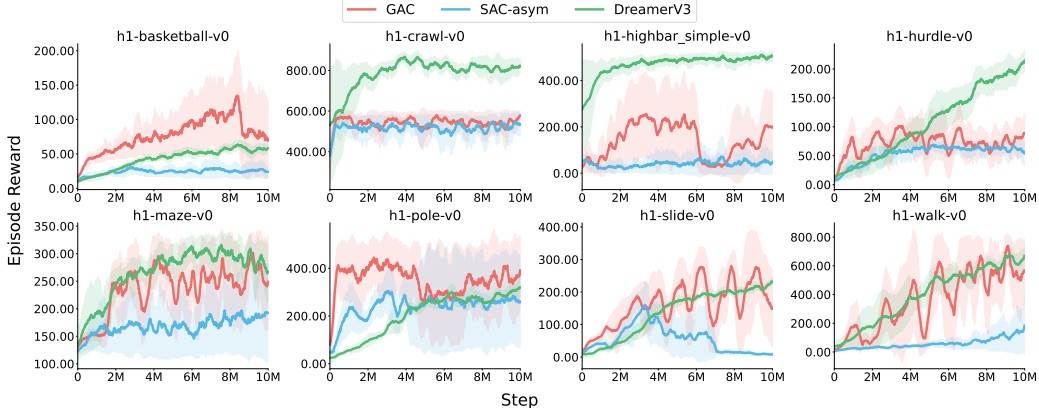

Figure 7: Performance comparison of GAC, SAC-asym and DreamerV3 on locomotion tasks in HumanoidBench.

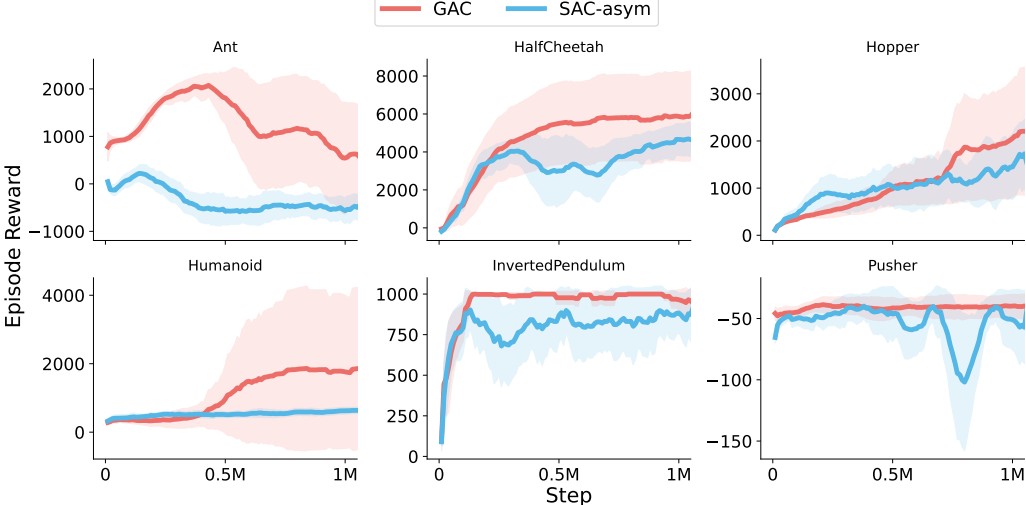

Figure 8: Performance comparison of GAC and SAC-asym on image-based tasks in MuJoCo.

- **Concentration**: A deck of cards is shuffled and spread out face down. The player flips two cards at a time face up, receiving a reward if the flipped cards match. The privileged information is a recorder that tracks all previous flipped cards.

- **MineSweeper**: The computer game MineSweeper, but the agent does not have access to the board. Each observation contains the position and number of adjacent mines to the last square "clicked" by the agent. The privileged information is a recorder that tracks all previous observations.

- **Repeat Previous**: At the first timestep, the agent receives one of four values and a remember indicator. Then it randomly receives one of the four values at each successive timestep without the remember indicator. The agent is rewarded for outputting the observation from some constant k timesteps ago. The privileged information is the exact value that should be output at each timestep.

**HumanoidBench** (Sferrazza et al., 2024). HumanoidBench is a high-dimensional simulated robotics benchmark featuring a humanoid robot equipped with dexterous hands, supporting a variety of challenging whole-body manipulation and locomotion tasks. For the locomotion tasks, the privileged information is defined as the velocity and angular velocity of all joints and objects. For the manipulation tasks, the privileged information is defined as the tactile feedback.

Table 7: Observation space data for the tasks chosen in Brax.

| Task | Original obs_dim | Partial obs_dim | Privileged obs_dim |
|---|---|---|---|
| Ant | 27 | 13 | 14 |
| HalfCheetah | 17 | 8 | 9 |
| Humanoid | 244 | 155 | 89 |
| HumanoidStandup | 244 | 155 | 89 |
| InvertedDoublePendulum | 8 | 5 | 3 |

Table 8: Observation space data for the tasks chosen in HumanoidBench.

| Task | Original obs_dim | Partial obs_dim | Privileged obs_dim |
|---|---|---|---|
| h1-basketball-v0 | 64 | 33 | 31 |
| h1-crawl-v0 | 51 | 26 | 25 |
| h1-highbar_simple-v0 | 51 | 26 | 25 |
| h1-hurdle-v0 | 51 | 26 | 25 |
| h1-maze-v0 | 51 | 26 | 25 |
| h1-pole-v0 | 51 | 26 | 25 |
| h1-slide-v0 | 51 | 26 | 25 |
| h1-walk-v0 | 51 | 26 | 25 |
| h1touch-basketball-v0 | 164 | 164 | 1344 |
| h1touch-bookshelf_hard-v0 | 308 | 308 | 1344 |
| h1touch-bookshelf_simple-v0 | 308 | 308 | 1344 |
| h1touch-insert_normal-v0 | 190 | 190 | 1344 |
| h1touch-insert_small-v0 | 164 | 164 | 1344 |
| h1touch-room-v0 | 229 | 229 | 1344 |
| h1touch-spoon-v0 | 167 | 167 | 1344 |
| h1touch-window-v0 | 171 | 171 | 1344 |

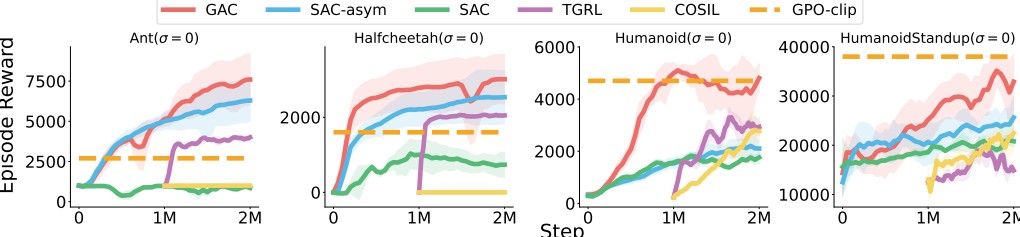

Figure 9: Performance comparison including COSIL.

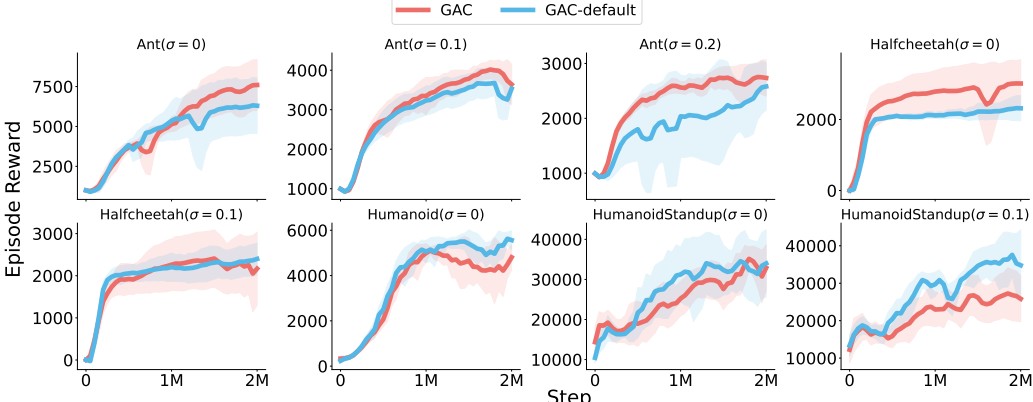

Figure 10: Performance comparison between tuned GAC and GAC with default target KL.

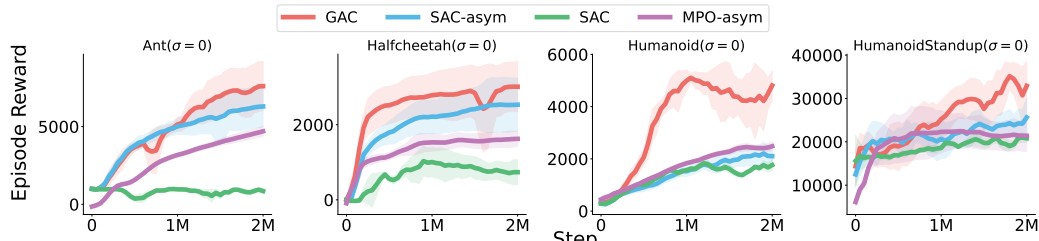

Figure 11: Performance comparison including asymmetric MPO.

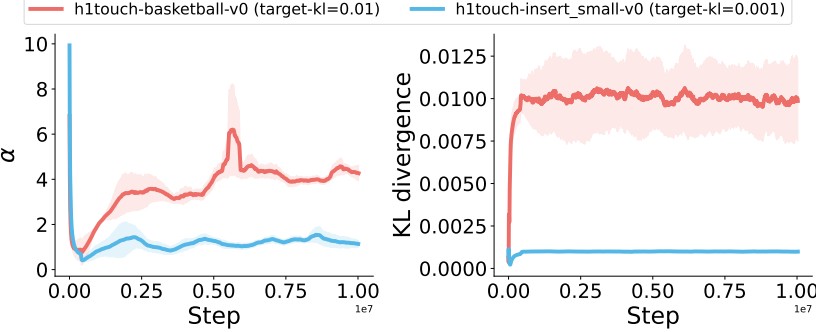

Figure 12: The temperature $\alpha$ and KL divergence between guider and learner during training.

