# OpenReview forum: "Guided Actor-Critic: Off-Policy Partially Observable Reinforcement Learning with Privileged Information"
_ICLR.cc/2026/Conference — Submitted to ICLR 2026_

### Official Review · Reviewer_sD5w · 2025-10-24

**Soundness:** 2
**Presentation:** 3
**Contribution:** 1
**Rating:** 4
**Confidence:** 5

**Summary:**

This paper proposes a novel off-policy reinforcement learning algorithm—Guided Actor-Critic (GAC)—for solving partially observable Markov decision processes (POMDPs) with privileged information. The method integrates a "guider" policy, which has access to the full state, and a "learner" policy, which only receives partial observations, within a jointly trained framework.

**Strengths:**

**Clear problem formulation and motivation.**:

The paper clearly articulates the challenges of leveraging privileged information during training in POMDPs and precisely identifies the limitations of the two existing paradigms—privileged policy learning and privileged value learning: the former may lead to suboptimal distillation, while the latter provides only indirect supervision.

**The core idea of GAC.**

Jointly optimizing the guider and learner under a KL-divergence constraint is relatively straightforward. It successfully extends the guiding principle of Guided Policy Optimization (GPO) from an on-policy setting to an off-policy one. The approach appears intuitively sound and practical.

**Weaknesses:**

**About Lemma and Method.**
In general partially observable Markov decision processes (POMDPs), the optimal policy under full observability (MDP) is typically unattainable—and often infeasible—under partial observability. This distinction is critical both theoretically and in practice.
In an MDP, the optimal policy \(\pi^*_{\text{MDP}}(a|s)\) is a function of the true state \(s\) and can make decisions directly based on complete state information.
- In a POMDP, the agent cannot observe the true state \(s\); instead, it must base decisions on the observation history \(\tau = (o_0, a_0, \dots, o_t)\), or equivalently, on the belief state \(b_t(s) = P(s_t = s \mid \tau)\). The optimal POMDP policy is thus \(\pi^*_{\text{POMDP}}(a|\tau)\) (or \(\pi^*(a|b_t)\)).
- Because the belief state generally contains strictly less information than the true state (i.e., \(I(b_t; s_t) < H(s_t)\)), the optimal return in a POMDP is usually **strictly lower** than that of the corresponding MDP:
  \[
  J^*_{\text{POMDP}} < J^*_{\text{MDP}}.
  \]

Therefore, **attempting to make a partially observable policy imitate the fully observable MDP-optimal policy is inherently misaligned**, as the latter may be unrealizable under partial observability and could even induce irrational behavior in the learner (e.g., "braking early" without perceptual evidence of an obstacle).

Two examples :

1. **Texas Hold’em Poker**:
   - A "God’s-eye-view" policy knows all players’ hole cards and can make theoretically optimal betting/folding decisions.
   - A real player, however, only knows their own cards and public information, and must infer opponents’ ranges. Imitating the God’s-eye policy would lead to overconfidence or incorrect inferences, reducing win rates.

2. **Occlusion in autonomous driving**:
   - A full-state policy might brake early because it "knows" a pedestrian is hidden behind a large vehicle.
   - A partially observable policy that blindly mimics this behavior—without sensory evidence—might brake unnecessarily in clear scenarios, compromising ride comfort or even causing rear-end collisions.
Given this, Theorem 3.4—which relies on that assumption—and the subsequent method described in Section 3.2 appear to be theoretically unsound.

I believe that in some partially observable environments (maybe contain some noisy observations), it is impossible to achieve the same policy as in the fully observable MDP. That said, I do not deny the empirical improvements the authors have achieved through engineering optimizations; however, the theoretical assumption underlying their approach seems somewhat inappropriate.

**About related works.**

I concern that "since supervision is provided indirectly through the RL objective, it may be less sample-efficient than methods that
leverage direct expert supervision."  In some environments, such as autonomous driving, direct policy distillation may fail to recover the optimal policy, whereas effective value-function guidance can succeed.  As Figure 1, SAC-asym outperforms TGRL in most cases.

**About Experiment.**

The experiments compare against SAC-asym (which, aside from the proposed method, appears to perform the best). However, this set of baselines seems relatively weak. Relevant approaches such as leverages privileged information in the value
function **UAAC**[1] (maybe an SAC-based variant), representation learning methods **Believer**[2] are not included in the comparison.

[1] Unbiased asymmetric reinforcement learning under partial observability. AAMAS 2022.

[2] Learning belief representations for partially observable deep rl. ICML 2023.

**Questions:**

Whether the additional training resources incurred by the two policies have been evaluated. I think the paper's approach leans toward practical computational efficiency,  it is necessary to evaluate the additional resources and time required.

---

> ### Author Response · Authors · 2025-11-25
>
> We thank the reviewers for their constructive feedback and thoughtful suggestions. We appreciate the time and effort dedicated to evaluating our work. The comments have helped us clarify our contributions and improve the presentation. Below, we address each concern in detail.
>
> > [**Weakness 1**] On Lemma 3.4, Optimality, and Partial Observability
>
> We thank the reviewer for raising this important point. We fully agree that in general POMDPs, the optimal fully observable MDP policy is typically **unattainable** for a partially observable agent.
>
> For this reason, **forcing a partially observable policy to imitate a fully observable one is indeed misaligned**, and may lead to clearly irrational behavior—exactly as the reviewer noted in their examples (poker and autonomous driving). This mismatch is also the **central limitation of existing teacher–student approaches** under asymmetric information: a privileged teacher may act on information the student cannot access, and blind imitation by the student leads to suboptimal or even unsafe policies.
>
> **Clarifying Our Claim**
>
> We would like to clarify that **our method does not attempt to make a partially observable learner match the fully observable MDP‐optimal policy**.
> Lemma 3.4 refers **only to optimality within the POMDP**, not within the fully observable MDP. Our theoretical result states that **the guider and the learner both converge to the same POMDP-optimal policy**. We do not claim superiority over other asymmetric RL methods in terms of global MDP optimality; all such methods are bounded by the same POMDP information structure.
>
> **How GAC Avoids the Reviewer’s Concern**
>
> The reviewer wrote:
>
> > “Attempting to make a partially observable policy imitate the fully observable MDP-optimal policy is inherently misaligned…”
>
> We fully agree. In fact, this is precisely the issue our method is designed to resolve.
> GAC **prevents the guider from exploiting privileged information arbitrarily** by enforcing a KL constraint (Eq. 3) that ties the guider’s policy to the learner’s policy. This restriction prevents the guider from collapsing to a fully observable (“God-mode”) strategy that the learner cannot imitate.
>
> Thus, unlike prior teacher–student methods, GAC ensures that the guider never drifts into an unreachable region of policy space, and the learner never attempts to imitate actions that are impossible to justify under partial observability.
>
> **Examples Supporting the Reviewer’s Concern**
>
> The reviewer’s two examples (poker and autonomous driving) accurately illustrate why naive privileged imitation fails. Previous asymmetric teacher–student approaches may indeed behave poorly in these settings, precisely because the teacher can condition on information the student does not have.
>
> In our paper, we highlight the same issue using the classical **Tiger Door** problem in Appendix B: a teacher who knows the true state may behave in a way that a partially observable student cannot rationalize, leading to degraded performance if the student imitates blindly.
>
> To further illustrate this effect and to show how GAC avoids it, we provide the following additional example:

---

> > ### Author Response · Authors · 2025-11-25
> >
> > Illustrative Example: How GAC Avoids Suboptimal Imitation under Partial Observability
> > --
> >
> > To clearly demonstrate how GAC avoids the suboptimality caused by blindly imitating a privileged teacher, we consider a simplified variant of the Tiger Door problem.
> >
> > **Problem Setup**
> >
> > There are two states
> > $$
> > S=\\{s_l,s_r\\}
> > $$
> > corresponding to the tiger being behind the left or right door.
> > The agent has two actions
> > $$
> > A=\\{a_l,a_r\\}
> > $$
> > opening the left or right door. The reward is:
> > $$
> > r(s_l,a_l)=1,r(s_r,a_r)=2,
> > $$
> > and zero otherwise.The tiger is behind either door with equal probability.
> >
> > A **privileged teacher** $\mu$, who observes the true state, can always choose the correct door:
> > $$
> > \mu(s_l)=[1,0],\mu(s_r)=[0,1],
> > $$
> > A student cannot distinguish $s_l$ from $s_r$ and must use a single policy $\pi(a∣o)$.
> > If the student naively imitates the teacher, it obtains
> > $$
> > \pi(a|o)=[0.5,0.5]
> > $$
> > whose expected reward is 0.75.
> > However, the **optimal partially observable policy** is $\pi(a|o)=[0,1]$ which yields reward 1.
> >
> > **How GAC Avoids This Failure**
> >
> > We initialize both teacher and student policies as uniform:
> > $$\mu(s_l)=\mu(s_r)=\pi(o)=[0.5,0.5],$$
> > so initially $D_{KL}(\mu,\pi)=0$.
> > The guided Q-values are simply the immediate rewards:
> > $$Q(s_l)=[1,0],Q(s_r)=[0,2]$$
> > Using the GAC update
> > $$\mu(\cdot|s) \propto \pi(\cdot) exp(Q(s,\cdot)/\alpha), \alpha=1$$
> > we obtaine
> > $$\mu(s_l)=[\frac{e}{1+e},\frac{1}{1+e}],\mu(s_r)=[\frac{1}{1+e^2},\frac{e^2}{1+e^2}]$$
> > The student then minimizes the KL divergence
> > $$\pi_{new}=argmin D_{KL}(\mu_{new}|\pi),$$
> > which, under a single undistinguished observation, reduces to averaging the two guider policies:
> > $$\pi_{new}=0.5\mu(s_l)+0.5\mu(s_r).$$
> > This yields
> > $$\pi(a_l|o)=\frac{0.5e}{1+e}+\frac{0.5}{1+e^2},
> > \pi(a_r|o)=\frac{0.5}{1+e}+\frac{0.5e^2}{1+e^2}
> > $$
> > Numerically $\pi=[0.425,0.575]$.
> > Thus GAC increases the probability of the better action $a_r$, even though the student does not observe the true state.
> > The student’s expected reward is 0.787, which is higher than the initial one.
> >
> > Thus, **GAC monotonically improves the student policy**, overcoming the failure of naive imitation and pushing the learner toward the optimal partially observable policy $[0,1]$.
> >
> >
> > > [**Weakness 2**] About related works. I concern that "since supervision is provided indirectly through the RL objective, it may be less sample-efficient than methods that leverage direct expert supervision." In some environments, such as autonomous driving, direct policy distillation may fail to recover the optimal policy, whereas effective value-function guidance can succeed. As Figure 1, SAC-asym outperforms TGRL in most cases.
> >
> > We thank the reviewer for the comment. Our original phrasing — that supervision provided indirectly through the RL objective may be less sample-efficient than direct expert supervision — is intentional and accurate.
> > Direct policy distillation often provides a strong, high-SNR learning signal and therefore can be more sample-efficient when the teacher’s behavior is realizable by the learner. Saying “may be less sample-efficient” therefore correctly captures this practical advantage of distillation.
> >
> > That said, this statement is not meant to imply that distillation is always superior.
> > In asymmetric settings — where the teacher acts on privileged information the learner does not have access to — naive distillation can be misaligned or even harmful. As the reviewer notes, in some tasks (including settings similar to those in Fig. 1) asymmetric value-based guidance (e.g., SAC-asym) recovers better student policies precisely because it does not force the learner to imitate unreachable, privileged behaviors. We agree with the reviewer that SAC-asym outperforms TGRL in many cases, and this observation supports the point we made.
> >
> > Importantly, asymmetric actor-critic methods also have downsides: they can be less sample-efficient, require careful value-estimator bootstrapping, and in some environments suffer from instability or slow credit assignment. These trade-offs motivate our design goal: leverage the complementary strengths of both approaches. Our method combines value-function guidance (robustness under asymmetry) with targeted policy distillation (sample-efficiency when the teacher is realizable), aiming to be effective across a broader range of environments than either approach alone.

---

> > > ### Author Response · Authors · 2025-11-25
> > >
> > > > [**Weakness 3**] About Experiment. The experiments compare against SAC-asym (which, aside from the proposed method, appears to perform the best). However, this set of baselines seems relatively weak. Relevant approaches such as leverages privileged information in the value function UAAC[1] (maybe an SAC-based variant), representation learning methods Believer[2] are not included in the comparison.
> > >
> > > The SAC-based variant of UAAC is **exactly** SAC-asym. For instance, in POPGym, the critic input includes the full trajectory history via a GRU—i.e.,
> > > $[current\ observation, last\ action, termination\ flag]_{1:t}$.
> > > —which corresponds to an unbiased asymmetric actor-critic setup. Therefore, our SAC-asym baseline already represents the SAC-based instantiation of UAAC.
> > >
> > > Regarding representation-learning approaches such as Believer, this paper does not aim to benchmark against representation learning or belief modeling frameworks. Believer is a modular architecture that separately learns a belief state $b(h_t)$ using VAE-based latent inference and then uses this belief as input to standard RL algorithms. This design is **orthogonal** to our focus: Believer can be combined with both GAC and all baselines (including SAC-asym). Since it provides a complementary memory/belief module rather than a competing policy-learning mechanism, we do not frame it as a direct baseline in this work.
> > >
> > > > [**Q1**] Whether the additional training resources incurred by the two policies have been evaluated. I think the paper's approach leans toward practical computational efficiency, it is necessary to evaluate the additional resources and time required.
> > >
> > > GAC includes two policy networks and four Q-networks, doubling the number of trainable parameters compared to SAC, and matching TGRL, which also uses two policies and four critics.
> > >
> > > Below we report the number of trainable parameters and training throughput (FPS) on Humanoid using a single RTX 4090:
> > >
> > > | Method | # Params  | FPS |
> > > | ------ | --------- | --- |
> > > | SAC    | 523,300   | 192 |
> > > | GAC    | 1,046,600 | 107 |
> > > | TGRL   | 1,046,600 | 111 |
> > >
> > > For off-policy algorithms, **sample efficiency** is typically the primary concern; wall-clock training time is less critical because off-policy methods amortize computation over many environment samples. If wall-clock time were the priority, one would typically adopt a PPO-based method such as GPO, which can reach comparable or even superior performance using less than 1/10 of the training time.

---

> > > ### Comment · Reviewer_sD5w · 2025-11-25
> > >
> > > My concern is: "Eq (1) states that The agent’s objective is to find an optimal policy π∗ that maximizes the expected return and Eq(3) formalizes the guider’s objective as a constrained reinforcement learning problem, the KL constraint is a small number ϵ  ". Lemma 3.4 assumes that the KL divergence converges to zero. In the provided example (e.g., the Tiger Door variant), it does not actually reach zero (e.g., the final KL remains positive, and the learner’s policy does not fully match the guider’s). I suspect that while the method empirically achieves improvements, maybe due to increased model capacity or the joint training dynamics. The theoretical proof (particularly the claim of KL → 0 and its implications for optimality) appears problematic.
> > >
> > > If my understanding is incorrect, please clarify.

---

> > > > ### Author Response · Authors · 2025-11-26
> > > >
> > > > 1. **Clarification for the Tiger Door Variant**
> > > >
> > > > First, note that **the KL we require to be zero in the theoretical analysis is Eq. (7), not Eq. (8)**.
> > > > In the Tiger Door example, we instantiate the guider as:
> > > >
> > > > $$\mu(\cdot|s) \propto \pi(\cdot) exp(Q(s,\cdot)/\alpha)$$
> > > >
> > > > which directly corresponds to the condition in **Eq. (7)** where the KL is minimized to zero.
> > > > The only reason the empirical KL between guider and learner (Eq. (8)) remains positive is because training has not converged. If we write down the converged solution:
> > > > $$\mu(s_l)=\mu(s_r)=\pi(o)=[0,1], Q(s_l)=[1,0],Q(s_r)=[0,2]$$
> > > > then the fixed‐point condition:
> > > > $$\mu(\cdot|s) \propto \pi(\cdot) exp(Q(s,\cdot)/\alpha)$$
> > > > holds exactly, and therefore: $D_{KL}(\pi,\mu)=0$.
> > > >
> > > > Thus, the Tiger Door example **does not contradict Lemma 3.4**; the algorithm simply had not reached the theoretical fixed point.
> > > >
> > > > 2. **The Role of Eq. (7) and Lemma 3.4**
> > > >
> > > > **Theorem 3.3 (Guided Policy Iteration)** already guarantees monotonic policy improvement under the constrained objective in Eq. (4). This theorem is the key result establishing optimality within the policy class.
> > > > Lemma 3.4 serves a different purpose: it characterizes **what the optimal policy for Eq. (4) looks like**. To obtain this closed-form solution, we assume **Eq. (7) achieves KL=0**.
> > > >
> > > > This assumption is theoretically mild because Eq. (7) is easy to satisfy: both sides of the KL share the same input and the same model capacity. In fact, KL=0 simply requires the guider network to output logits of the form:
> > > > $$\log\pi(a|o_t)+Q(s_t,a)/\alpha+C(s_t)$$
> > > > which it can represent.
> > > >
> > > > In practice, we do not separately optimize Q to convergence and then optimize the policy to convergence. Instead, Q and the policies are updated interleaved, once per iteration—just like in SAC.
> > > > Under such alternating updates, KL in Eq. (7) rarely reaches zero at intermediate iterations, even though the fixed point requires KL=0. This is expected and not problematic.
> > > >
> > > > 3. **About Eq. (8): Why Empirical KL ≠ 0 Is Not a Problem**
> > > >
> > > > Although theory also allows Eq. (8)’s KL to be zero at convergence, in practice it settles at a small positive value.
> > > >
> > > > The situation is almost identical to SAC:
> > > > - SAC’s soft policy iteration is optimal **under the entropy-regularized objective**.
> > > > - To guarantee optimality under **the original RL objective**, one must anneal $\alpha$ to 0.
> > > > - But in practice, SAC does **not** anneal $\alpha$ to zero because a small $\alpha$ helps exploration without harming performance.
> > > >
> > > > Analogously:
> > > > - GAC is optimal **under the constrained objective** in Eq. (4).
> > > > - To guarantee optimality under the **original RL objective**, KL in Eq. (8) should ideally go to zero.
> > > > - But in practice, keeping a small KL is beneficial: it allows the guider to continue exploiting privileged information and collecting better trajectories.
> > > >
> > > > Indeed, Fig. 12 in the revised paper shows that we can fully control the target KL if desired. One could anneal KL to 0 to ensure full theoretical consistency, but practically we find this unnecessary.

---

> ### Comment · Reviewer_sD5w · 2025-11-26
>
> We explicitly define:
> - $\mu$ as a ***global*** policy — one that makes optimal decisions using full knowledge of the environment state:
>   $\mu = \mu (a \mid s)$, i.e., it **depends on the true state $s$**, but is independent of the observation $o$;
> - $\pi^*$ as an ***observation-based*** (local) policy:
>   $\pi = \pi (a \mid o)$  or more generally $\pi(a \mid \tau)$ , operating solely on partial observations.
>
> My concern centers on **Lemma 3.4**, specifically **Equation (11)** (not Eq. (7) or Eq. (8)).
> Lemma 3.4 assumes that the KL divergence converges to zero **and** that *both* $\mu^\*$ and $\pi^\*$ converge to the *same* optimal policy.
>
> However, this appears inconsistent with the policy definitions above (Poker Driving Tiger and so on).
> In the *Clarification for the Tiger Door Variant*, the authors state a converged solution where:
> $$
> \mu(s_\ell) = \mu(s_r) = \pi(o) = [0, 1].
> $$
> But if $\mu$ is a *privileged*, fully observable ("teacher") policy, it **should** distinguish between $s_\ell$ and $s_r$, i.e., $\mu(s_\ell) \ne \mu(s_r)$ in general. Enforcing $\mu(s_\ell) = \mu(s_r)$ effectively reduces $\mu$ to an *observation-based* policy, contradicting its definition as a *state-conditional global policy*.
>
> In other words:
> - The example used in the clarification assumes a *fixed* $\mu$ that does **not** exploit privileged information (i.e., it behaves identically in both states),
> - But Lemma 3.4 claims convergence to a *common* optimal policy $\mu^{\*} = \pi^{\*}$, implicitly requiring $\mu^{\*}(s_l) = \mu^{\*}(s_r) = \pi^{\*}(o)$.
>
> **This is not an issue about KL divergence values**, it may be a **fundamental mismatch with the definitions of the policies** themselves.
>
> To reiterate:
> - My main concern is Eq. (11) in Lemma 3.4, where the optimal guider and learner policies are asserted to coincide.
> - For a truly privileged $\mu^\*$ (i.e., $\mu^\*(a \mid s)$), this equality can only hold in degenerate POMDPs where the optimal action is *state-independent* — which is generally **not** the case (e.g., Tiger Door, Poker, occluded driving).
>
> If my understanding is incorrect, please clarify.

---

> ### Author Response · Authors · 2025-11-26
>
> Thank you for your timely reply.
>
> First, we would like to clarify a key point of notation.
> In line 90, our definition of $s$ is **broader than the true environment state**. In practice, the guider receives: _partial observations + privileged information_.
> Therefore, the guider’s input space **contains** the learner’s input.
> This means the guider can choose to ignore the privileged information and act solely based on the partial observation, which makes it possible for the guider’s optimal policy to coincide with the learner’s policy.
>
> **Why $\mu$ does not necessarily converge to the full-state optimal policy**
>
> In standard RL, a policy that receives privileged information indeed tends to exploit it.
> However, GAC is not different.
>
> The guider $\mu$ is optimized under a KL constraint w.r.t. the learner (Eq. (3)), enforced through Eq. (7) and the corresponding supervised loss Eq. (13): minimize $\alpha D_{KL}(\mu,\pi) - Q$.
>
> Thus, the guider’s update direction is not _maximize Q purely using full state_, but instead a Q-modulated version of the learner policy (Eq. (7)).
> Therefore, the guider is not free to exploit all available privileged information.
> The constrained optimization pulls $\mu$ towards the learner’s policy class.
>
> **Tiger Door Variant: Why the Guider and Learner Both Converge to [0,1]**
>
> To explicitly illustrate that the guider does not converge to a “fully privileged full-state optimal policy”, we show the tabular iteration starting from a uniform policy (with $\alpha=2$ for faster convergence):
>
> |Iter | $\mu(s_l)$ | $\mu(s_r)$  | $\pi$ | $Q(s_l)$ |$Q(s_r)$|
> | ------ | --------- | --- |--- |--- |--- |
> |0|[0.5 0.5]| [0.5 0.5]| [0.5 0.5]| - | - |
> |1|[0.622 0.378]| [0.269 0.731]| [0.446 0.554]| [ 1. -0.]| [-0.  2.]|
> |2|[0.57 0.43]| [0.228 0.772]| [0.399 0.601]| [ 0.937 -0.063]| [-0.066  1.934]|
> |3|[0.523 0.477]| [0.196 0.804]| [0.36 0.64]| [ 0.941 -0.059]| [-0.066  1.934]|
> |4|[0.481 0.519]| [0.171 0.829]| [0.326 0.674]| [ 0.945 -0.055]| [-0.064  1.936]|
> |5|[0.444 0.556]| [0.151 0.849]| [0.297 0.703]| [ 0.949 -0.051]| [-0.061  1.939]|
> |6|[0.411 0.589]| [0.135 0.865]| [0.273 0.727]| [ 0.952 -0.048]| [-0.058  1.942]|
> |7|[0.382 0.618]| [0.121 0.879]| [0.252 0.748]| [ 0.956 -0.044]| [-0.055  1.945]|
> |8|[0.357 0.643]| [0.11 0.89]| [0.233 0.767]| [ 0.959 -0.041]| [-0.053  1.947]|
> |9|[0.334 0.666]| [0.101 0.899]| [0.217 0.783]| [ 0.962 -0.038]| [-0.05  1.95]|
> |10|[0.314 0.686]| [0.093 0.907]| [0.204 0.796]| [ 0.964 -0.036]| [-0.048  1.952]|
> |...|...|...|...|...|...|
> |100|[0.033 0.967]| [0.008 0.992]| [0.02 0.98]| [ 0.997 -0.003]| [-0.005  1.995]|
>
> All three distributions—$\mu(s_l)$, $\mu(s_r)$ and $\pi$—converge toward [0,1].

---

### Official Review · Reviewer_8jcT · 2025-10-28

**Soundness:** 3
**Presentation:** 3
**Contribution:** 2
**Rating:** 4
**Confidence:** 4

**Summary:**

The paper proposes Guided Actor–Critic (GAC), an off-policy algorithm for RL with training-time privileged information under partial observability. The key idea is a guided policy iteration scheme that co-trains a guider policy and a learner policy under a KL alignment constraint. Empirically, GAC is benchmarked on Brax with noisy observations, POPGym memory tasks, and HumanoidBench manipulation with tactile signals available only during training. The authors report better sample efficiency and final returns than asymmetric SAC, TGRL/teacher-student, RMA, and on-policy GPO-clip.

**Strengths:**

1.The paper tackles the important and practical problem of sample-efficient learning in partially observable environments using training-only privileged information.

2.The paper is well-written and situates itself perfectly within the existing literature, with a clear motivation and a thorough related work review.

**Weaknesses:**

1.The final GAC algorithm is extremely complex, requiring six separate neural networks (a guider policy, a learner policy, two guider Q-networks, and two learner Q-networks) and five distinct loss functions (two Q-losses, two policy losses, one $\alpha$-loss). This high complexity is a barrier to reproduction and adoption.

2.The tabular analysis suggests a KL-aligned learner (distillation-like), but the practical method adds an auxiliary RL loss for the learner and separate learner critics. Ablations indicate the auxiliary learner RL loss is important for performance; separate Qs are often helpful but not universally critical. This weakens the practical relevance of the “pure distillation” narrative.

3.Although the paper formally relates GAC to MPO, it provides no empirical MPO-style baseline (with/without privilege), leaving it unclear whether gains stem from asymmetry + guidance or from a generic KL-regularized update.

**Questions:**

See Weaknesses

---

> ### Author Response · Authors · 2025-11-25
>
> We thank the reviewers for their constructive feedback and thoughtful suggestions. We appreciate the time and effort dedicated to evaluating our work. The comments have helped us clarify our contributions and improve the presentation. Below, we address each concern in detail.
>
> > [**Weakness 1**] The final GAC algorithm is extremely complex, requiring six separate neural networks (a guider policy, a learner policy, two guider Q-networks, and two learner Q-networks) and five distinct loss functions (two Q-losses, two policy losses, one
> -loss). This high complexity is a barrier to reproduction and adoption.
>
> While GAC uses more networks than standard SAC, this increase is natural because GAC contains two agents—a guider and a learner. Consequently, the number of networks and loss functions simply doubles, just as in other teacher–student frameworks. For example, TGRL, a recent teacher–student method, also maintains two policies and four Q-networks, with two Q-losses and two policy-losses.
>
> Despite the apparent complexity, **implementing GAC is straightforward**.
> The guided policy iteration in GAC closely mirrors the soft policy iteration used in SAC. In practice, extending any standard SAC implementation requires only minimal, mostly mechanical changes. For instance, starting from CleanRL’s SAC code, GAC can be implemented with roughly **40 additional lines**, involving:\
> (1) defining the extra guider/learner networks,\
> (2) duplicating the Q-update and policy-update steps,\
> (3) inserting the guider–learner KL term into both actor losses.
>
> Compared to other teacher–student or off-policy KL-based methods such as TGRL or MPO, **GAC is actually simpler to implement**, with fewer custom components and no need for complicated constrained optimization.
>
>
> > [**Weakness 2**] The tabular analysis suggests a KL-aligned learner (distillation-like), but the practical method adds an auxiliary RL loss for the learner and separate learner critics. Ablations indicate the auxiliary learner RL loss is important for performance; separate Qs are often helpful but not universally critical. This weakens the practical relevance of the “pure distillation” narrative.
>
> We agree that GAC is **not a pure distillation method**. The motivation for adding the auxiliary learner RL loss is straightforward: the replay buffer is generated by a policy that is **already very close to the learner**, so applying RL directly on the learner allows it to exploit this data more effectively. This aligns with modern teacher–student and distillation frameworks, where auxiliary RL objectives are commonly used to improve stability and performance (e.g., [1], [2], [3]). Pure distillation alone is generally insufficient in challenging POMDP settings, and our ablations consistently show that combining guided learning with learner-side RL is benifical.
>
> [1] Tgrl:An algorithm for teacher guided reinforcement learning.
>
> [2] Leveraging fully observable policies for learning underpartial observability.
>
> [3] Bridging the imitationgapby adaptive insubordination.
>
> > [**Weakness 3**] Although the paper formally relates GAC to MPO, it provides no empirical MPO-style baseline (with/without privilege), leaving it unclear whether gains stem from asymmetry + guidance or from a generic KL-regularized update.
>
> We did not originally include MPO because it is generally **less competitive and less widely adopted** than SAC. As reported in the ACME paper [4], MPO consistently underperforms SAC on standard continuous-control benchmarks such as Ant and Humanoid. In the revised version, we have added an **asymmetric-MPO** baseline (Figure 11). The results follow the same trend: asymmetric-MPO is notably weaker than asymmetric-SAC.
>
> [4] Acme: A Research Framework for Distributed Reinforcement Learning.

---

### Official Review · Reviewer_Rgrc · 2025-10-31

**Soundness:** 3
**Presentation:** 2
**Contribution:** 2
**Rating:** 4
**Confidence:** 4

**Summary:**

The paper proposes Guided Actor-Critic (GAC) for POMDPs that leverages training-time privileged information. The method co-trains a privileged guider policy $\mu(a\mid s)$ and a partially observable learner policy $\pi(a\mid o)$ via guided policy iteration: (i) guided evaluation applies a modified Bellman backup that includes a KL term between $\mu$ and $\pi$; (ii) guided improvement updates $\mu$ toward a $Q$-weighted version of $\pi$ and then distills $\mu$ into $\pi$ via KL. The paper provides tabular convergence and a practical deep RL algorithm with automatic KL temperature and clipped double Q. On Brax with observation noise, POPGym memory tasks, and HumanoidBench where tactile signals serve as privileged info, GAC improves sample efficiency and final performance over asymmetric SAC, TGRL, and GPO-clip, and analyzes failure modes at high noise and target-KL sensitivity.

**Strengths:**

Unified use of privileged info: combines privileged policy learning and privileged value learning in a single off-policy framework, improving sample efficiency.

Clear algorithmic structure: guided evaluation and improvement are well specified; the learner receives direct supervision from the guider plus RL signals.

Theory plus practice: convergence in the tabular case and a practical deep variant with automatic KL control and clipped double Q.

Broad benchmarks: consistent gains on Brax (noisy partial observations), POPGym (memory), and HumanoidBench (tactile privileged info).

Insightful analysis: identifies a high-noise failure case and studies the effect of the target KL range.

**Weaknesses:**

Function-approximation theory gap: convergence results are tabular; there is no error-propagation or stability analysis with neural critics and replay.

Reliance on value quality: the guider update uses $Q$-weighted reweighting; bias in $Q$ under partial observability may misguide $\mu$ and $\pi$.

Target-KL tuning sensitivity: despite auto-temperature, performance can degrade for extreme target KL and very noisy observations.

Baselines: more competitive off-policy privileged baselines (e.g., MPO-style privileged variants) would sharpen the study.

Ablations: limited diagnostics on how much gains come from KL supervision vs learner RL, and on guider capacity assumptions.

**Questions:**

$Q$ robustness: have you tried ensembles or uncertainty-aware weighting in the $Q$-weighted improvement of $\mu$ to reduce bias in early training

Guider capacity: how sensitive are results to the architecture of $\mu$ and to the assumption that $\Pi_{\mu}$ can drive the KL in the improvement step near zero

Replay coupling: since data are collected by $\mu$ while $\pi$ is supervised, do you observe distribution-shift issues in the learner critic If so, any remedies beyond target networks

Memory modeling: on POPGym, what recurrent modules are used for $\mu$ and $\pi$ Are the architectures symmetric If not, how much does this matter

HumanoidBench tactile: can you quantify how often $\pi$ imitates $\mu$ vs deviates when tactile info was critical for $\mu$ but unavailable to $\pi$

---

> ### Author Response · Authors · 2025-11-25
>
> We thank the reviewers for their constructive feedback and thoughtful suggestions. We appreciate the time and effort dedicated to evaluating our work. The comments have helped us clarify our contributions and improve the presentation. Below, we address each concern in detail.
>
> > [**Weakness 1**] Function-approximation theory gap: convergence results are tabular; there is no error-propagation or stability analysis with neural critics and replay.
>
> We agree. This limitation is shared by most algorithmic RL papers—including SAC and MPO—which also provide tabular or idealized convergence arguments without fully addressing stability under deep function approximation and replay. Our theoretical setting is consistent with this established practice. A deeper analysis would require a dedicated theory paper.
>
> > [**Weakness 2**] Reliance on value quality: the guider update uses Q-weighted reweighting; bias in Q under partial observability may misguide $\mu$ and $\pi$.
>
> Yes. However, this reliance on the quality of the learned Q-function is inherent to all off-policy actor–critic methods, including SAC and TD3. Our explicit dependence appears only in Eq. 13, which plays the same role as the Q-based improvement step in those baselines.
>
> > [**Weakness 3**] Target-KL tuning sensitivity: despite auto-temperature, performance can degrade for extreme target KL and very noisy observations.
>
> Indeed, but this sensitivity is unavoidable for any method when hyperparameters are pushed to extremes. For example, SAC’s “default” target entropy varies substantially across domains (e.g., −0.89log(1/action_dim) for discrete POPGym and −0.5action_dim for continuous Brax), and using an inappropriate target entropy can severely degrade performance.
> Similarly, GAC can adopt a domain-specific default target-KL.
> We report results with target-KL = 0.001 on Brax in Fig. 10 of the revised paper and observe that GAC is not particularly sensitive to this value.
>
> > [**Weakness 4**] Baselines: more competitive off-policy privileged baselines (e.g., MPO-style privileged variants) would sharpen the study. Also, recurrent baselines (e.g., recurrent PPO [2]) are natural baselines for memory-heavy tasks.
>
> We did not originally include MPO because it is generally **less competitive and less widely adopted** than SAC. As reported in the ACME paper [1], MPO consistently underperforms SAC on standard continuous-control benchmarks such as Ant and Humanoid. In the revised version, we have added an **asymmetric-MPO** baseline (Figure 11). The results follow the same trend: asymmetric-MPO is notably weaker than asymmetric-SAC.
>
> Regarding recurrent PPO, the comparison is not directly aligned with our setting.
> PPO is **on-policy**, whereas GAC and all our baselines are **off-policy**, making sample efficiency differ by orders of magnitude. This gap is particularly evident in memory-heavy tasks.
> For example, on POPGym’s AutoencodeHard task, GAC reaches a return of −0.3 at **2M** steps, while PPO (reported in the POPGym paper) and asymmetric-PPO (reported in the GPO paper) both plateau around −0.45 even after **15M** steps. Thus, recurrent PPO is substantially less sample-efficient and does not provide a competitive baseline in our experimental regime.
>
> [1] Acme: A Research Framework for Distributed Reinforcement Learning.
>
> > [**Weakness 5**] Ablations: limited diagnostics on how much gains come from KL supervision vs learner RL, and on guider capacity assumptions.
>
> We provide this ablation in Appendix F.3: GAC_wo_Q removes the learner’s RL loss and isolates the effect of KL supervision.
>
> Regarding guider capacity, we do not have a direct metric for determining whether the guider is “sufficiently large.” One could increase the guider’s hidden size, but any performance change would confound increased capacity with increased parameter count, making it difficult to attribute gains specifically to capacity limits.
>
> Both theoretically and empirically, neural networks are universal approximators. As long as the guider is not significantly smaller than the learner and Q-function, its capacity should be sufficient for minimizing the KL. We therefore do not observe guider capacity to be a bottleneck in practice.

---

> > ### Author Response · Authors · 2025-11-25
> >
> > > [**Q1**] Q robustness: have you tried ensembles or uncertainty-aware weighting in the Q-weighted improvement of $\mu$ to reduce bias in early training.
> >
> > We have not, in order to preserve fairness with baselines, which also rely on single/double Q critics for policy improvement. Ensemble critics would likely improve performance for **all** methods, including ours, but GAC does not uniquely require such techniques.
> >
> > > [**Q2**] Guider capacity: how sensitive are results to the architecture of $\mu$ and to the assumption that $\Pi_{\mu}$ can drive the KL in the improvement step near zero.
> >
> > In practice, results are not sensitive as long as the guider $\mu$ is no smaller or simpler than the learner or Q-functions. Under universal function approximation, a sufficiently expressive $\mu$ can always realize the optimum of the improvement step. For instance, a network that outputs logits proportional to
> > $$\log\pi(a|o_t)+Q(s_t,a)/\alpha+C(s_t)$$
> > (for some normalization term C) would drive the KL to zero.
> >
> > The only case where exact KL minimization becomes infeasible is when the guider network has **strictly lower** expressive capacity than the learner or critic.
> >
> > > [**Q3**] Replay coupling: since data are collected by $\mu$ while $\pi$ is supervised, do you observe distribution-shift issues in the learner critic. If so, any remedies beyond target networks
> >
> > We do not observe significant distribution shift: evaluation uses $\pi$, and poor critic generalization would manifest directly in its returns.
> > If needed, one could mitigate this issue by annealing the behavior policy from $\mu$ to a mixture of $\mu$ and $\pi$, similar to DAgger. However, our current GAC implementation is not directly compatible with such annealing (as discussed in our response to Reviewer KRDz), and empirically we have not found it necessary.
> >
> > > [**Q4**] Memory modeling: on POPGym, what recurrent modules are used for $\mu$ and $\pi$. Are the architectures symmetric If not, how much does this matter
> >
> > For POPGym, $\mu$ and $\pi$ use the **same architecture**: two layer MLP with a single-layer GRU. All experiments—including POPGym—use symmetric architectures for the two policies. As discussed above, GAC only requires the guider to be no smaller than the learner; thus architectural symmetry is sufficient, and differences in recurrence design do not affect applicability.
> >
> > > [**Q5**] HumanoidBench tactile: can you quantify how often $\pi$ imitates $\mu$ vs deviates when tactile info was critical for $\mu$ but unavailable to $\pi$.
> >
> > GAC explicitly constrains the KL divergence between $\pi$ and $\mu$, so the learner is encouraged to imitate the guider throughout training. The degree of imitation can be quantified directly by tracking the KL divergence.
> > Figure 12 of the revised paper reports this metric. The KL between $\pi$ and $\mu$ stays close to the target-KL during training, indicating that the two policies remain consistently aligned even when tactile information is unavailable to $\pi$.
> >
> > While one could inspect per-action differences, such fine-grained comparisons are difficult to interpret. The KL trajectory provides a clearer and more meaningful measure of how closely $\pi$ follows $\mu$ under GAC.

---

> > > ### Comment · Reviewer_Rgrc · 2025-11-26
> > > **Response to authors**
> > >
> > > Thank you for your responses. Most of my concerns are solved by the authors, making the results and conclusions more convincing. Consequently, I updated the overall score from 4 to 6.

---

### Official Review · Reviewer_JrGq · 2025-11-01

**Soundness:** 2
**Presentation:** 3
**Contribution:** 2
**Rating:** 4
**Confidence:** 3

**Summary:**

The paper provides an approach for combining asymmetric actor-critic and teacher-student policies for leveraging privileged information in POMDPs. The paper presents experiments on various benchmarks and provides analysis that demonstrates the convergence of their actor-critic approach.

**Strengths:**

- Combining asymmetric actor-critic with a teacher-student approach. To provide a stronger supervision signal to the actor.
- Adding the divergence between the teacher and student policy to the value function backup seems like it might lead to the teacher policy taking actions that keep the agent in regions where the student agent can imitate the teacher. It appears to be a novel idea, although related to [1] and [2].

[1] Nguyen, Hai, et al. "Leveraging fully observable policies for learning under partial observability." arXiv preprint arXiv:2211.01991 (2022).

[2] Messikommer, Nico, et al. "Student-informed teacher training." arXiv preprint arXiv:2412.09149 (2024).

**Weaknesses:**

- It is not entirely clear why the provided method is superior to other similar approaches, for example, [1], which also leverages a teacher policy alongside a symmetric actor-critic approach.
- The epsilon variable needs to be tuned as well; it's not very clear how sensitive the method is to such a variable
- no ablation studies

**Questions:**

- Can the authors provide ablation studies of the different parts of their algorithm? It's difficult to disentangle the effect of each module as the paper currently stands
- Can the authors compare their algorithms with other methods combining teacher/student policies and asymmetric actor-critic, such as [1] ?
- Why can't your approach be off-policy?

I would be willing to raise my score if the authors can provide the required experiments.

---

> ### Author Response · Authors · 2025-11-25
>
> We thank the reviewers for their constructive feedback and thoughtful suggestions. We appreciate the time and effort dedicated to evaluating our work. The comments have helped us clarify our contributions and improve the presentation. Below, we address each concern in detail.
>
>
> > [**Weakness 1 & Q2**] It is not entirely clear why the provided method is superior to other similar approaches, for example, [1], which also leverages a teacher policy alongside a symmetric actor-critic approach. Can the authors compare their algorithms with other methods combining teacher/student policies and asymmetric actor-critic, such as [1]?
>
> 1. **Comparison to other teacher–student methods**
>
> The core advantage of **GAC** is that it avoids the **suboptimality issues** that classical teacher–student frameworks encounter in asymmetric settings. A canonical example is TigerDoor, illustrated in Appendix B.
>
> There are two major categories of teacher–student approaches:
>
> - **Traditional imitation-learning–based methods**.
> These methods can be _strictly suboptimal_ in tasks like TigerDoor because the teacher’s privileged policy exploits information unavailable to the learner, producing supervision the learner cannot follow.
>
> - **Distillation-style approaches (IL loss + RL loss)**.
> These methods can eventually overcome suboptimality by effectively **discarding teacher supervision** and relying on the RL loss alone. However, this also means they cannot effectively leverage privileged information.
>
> **GAC outperforms both categories** because it simultaneously uses (i) privileged teacher (guider) supervision and (ii) RL signals, **without being suboptimal**.
> The key reason is that classical teacher–student methods train an _unconstrained teacher_ that may become “over-optimal” (again, in TigerDoor), producing guidance the learner cannot imitate.
> **GAC explicitly constrains the guider via a KL bound**, ensuring the teacher remains _imitable_, which enables consistent and effective privileged supervision.
>
> 2. **Comparison to other asymmetric actor–critic methods**
>
> While asymmetric actor–critic does not suffer from teacher–student suboptimality, it leverages privileged information **only in the value function**, providing _indirect guidance_ to the actor.
>
> By contrast, GAC:
>
> - uses a privileged guider policy to provide direct supervision to the learner, and
>
> - collects trajectories using the privileged guider, whereas asymmetric AC collects trajectories using the partially observable actor.
>
> This yields more effective exploitation of privileged information in both learning signals and data collection, enabling GAC to outperform asymmetric actor–critic baselines.
>
> 3. **Comparison to COSIL [1]**
>
> We did not directly compare to COSIL because the TGRL paper reports that COSIL **underperforms TGRL**; thus, by comparing against TGRL, GAC implicitly outperforms COSIL.
>
> COSIL also has two inherent limitations:
>
> - **Theoretical limitation**:
> Like other teacher–student methods, COSIL cannot effectively utilize teachers that are too optimal or suboptimal. This is discussed in Section 6 of the COSIL paper, where COSIL even underperforms SAC.
>
> - **Practical limitation**:
> COSIL requires extensive hyperparameter tuning, especially the predefined KL target between teacher and student.
> In highly asymmetric settings (e.g., Brax), selecting this KL target is difficult.
>
> In our experiments, while GAC only needs a simple choice of KL in {0.01, 0.001, 0.005}, we performed a broad hyperparameter sweep for COSIL over
> {0.1, 0.2, 0.4, 0.8, 1, 1.2, 1.4, 1.8, 2},
> and obtained valid results for only two tasks (Humanoid, HumanoidStandup), as shown in Fig. 9 of the revised PDF.
> A KL target that is too large renders teacher supervision ineffective; too small causes the coefficient α to diverge.
>
> [1] Nguyen, Hai, et al. "Leveraging fully observable policies for learning under partial observability."

---

> > ### Author Response · Authors · 2025-11-25
> >
> > > [**Weakness 2**] The epsilon variable needs to be tuned as well; it's not very clear how sensitive the method is to such a variable.
> >
> > To assess sensitivity, we performed experiments on all Brax tasks using a **single default value** $\epsilon=0.001$.
> > Results (Fig. 10 in the revised PDF) compare GAC-default (untuned $\epsilon$) with GAC (tuned).
> > The performance gap is small across all tasks, demonstrating that **GAC is not sensitive to $\epsilon$**.
> >
> > > [**Weakness 3 & Q1**] no ablation studies. Can the authors provide ablation studies of the different parts of their algorithm? It's difficult to disentangle the effect of each module as the paper currently stands.
> >
> > We include ablations in Appendix F.3 evaluating:
> >
> > - **GAC_share**: guider and learner share the Q-function
> >
> > - **GAC_wo_Q**: learner removes the RL loss
> >
> > These ablations correspond exactly to the only two degrees of freedom in the GAC design: (i) whether the guider and learner maintain separate critics, and (ii) whether the learner is trained with RL objectives.
> >
> > Importantly, the core components of GAC—Eq. (12) for the critic, Eq. (13) for the guider, and Eq. (14) for the learner—form the **minimal functional set** required for the algorithm to operate.
> > Any additional removal would break the algorithm entirely (e.g., removing the guider update, the learner update, or the Q-function update makes training ill-defined), and therefore would not constitute a meaningful ablation.
> >
> > > [**Q3**] Why can't your approach be off-policy?
> >
> > **GAC is off-policy**.
> > If the intended question is “Why can your approach be off-policy?” or “Why can’t it be on-policy?”, we are happy to clarify.

---

### Official Review · Reviewer_KRDz · 2025-11-09

**Soundness:** 3
**Presentation:** 4
**Contribution:** 3
**Rating:** 8
**Confidence:** 3

**Summary:**

The paper proposes a policy optimization algorithm that leverages privileged information at training time. Specifically, a "guider" policy with access to privileged information, is trained to find high rewards while being softly constrained to the unprivileged learner policy. The learner policy is trained to imitate the guider. The paper establishes some theoretical backing for this method, and evaluate it on several benchmarks where it outperforms other privileged information baselines.

**Strengths:**

- Very nicely written and presented. The paper does a good job of setting up the problem, discussing the body of related works, explaining the method and necessary theory, and then showing the strengths and limitations of the method
- theoretical backing seems solid, although with the caveat that the KL divergence is actually met
- empirical section is strong. the authors compare against a variety of privileged info baselines, and in different domains.

**Weaknesses:**

I don't see any major weaknesses, but there are some minor ones to round out the paper.

- Discussion of "on-policy-ness" of using a guided policy to generate data for the learner. I could see in early stages, the privileged guider is useful. But eventually, it seems like a good idea to anneal to just the learner policy to generate data to avoid off policy issues, even if the guider is being softly constrained to act like the learner. Could there potential convergence issues or unstable training dynamics since the guider is likely always going to be a bit "off policy" wrt learner?
    - In Section 4.4, there is an experiment where KL divergence fails to converge. But what if you just train the learner on a convex combination of the guider and learner state action distribution, and anneal towards the learner over time?

- It would be nice to have some more task-specific  qualitative analysis / interesting behaviors / visual figures in the experimental section.
- Missing some references. The seminal work is Vapnik's SVM+[0]. Missing some privileged model-based rl works as well. Would like to see Vapnik properly cited in main work, and MBRL works could be supplemental in Appendix A.

0. A new learning paradigm: Learning using privileged information
1. The Wasserstein Believer: Learning Belief Updates for Partially Observable Environments through Reliable Latent Space Models (ICLR24)
2. Informed POMDP: Leveraging Additional Information in Model-Based RL (RLC25)
3. Privileged Sensing Scaffolds Reinforcement Learning (ICLR24)
4. TWIST: Teacher-Student World Model Distillation for Efficient Sim-to-Real Transfer (ICRA24)

**Questions:**

See weaknesses above. Overall I am fairly positive about this paper, although this is subject to change depending on the author's response and other reviewers' concerns.

---

> ### Author Response · Authors · 2025-11-25
>
> We thank the reviewers for their constructive feedback and thoughtful suggestions. We appreciate the time and effort dedicated to evaluating our work. The comments have helped us clarify our contributions and improve the presentation. Below, we address each concern in detail.
>
> > [**Weakness 1**] Discussion of "on-policy-ness" of using a guided policy to generate data for the learner. I could see in early stages, the privileged guider is useful. But eventually, it seems like a good idea to anneal to just the learner policy to generate data to avoid off policy issues, even if the guider is being softly constrained to act like the learner. Could there potential convergence issues or unstable training dynamics since the guider is likely always going to be a bit "off policy" wrt learner?
>
> > In Section 4.4, there is an experiment where KL divergence fails to converge. But what if you just train the learner on a convex combination of the guider and learner state action distribution, and anneal towards the learner over time?
>
> Theoretically, guided policy iteration ensures that the guider and learner converge to the **same** policy, so annealing the behavioral policy is not required in the ideal case.
> In practice, however, there will always be some residual divergence between the guider and learner. Thus, **annealing the behavioral policy toward the learner** (as in DAGGER-style schemes) could indeed help mitigate distribution shift, and we agree that this is a reasonable modification.
>
> However, there is a practical complication related to **exploration**.
> Since guided policy iteration ultimately converges to **deterministic** policies, the learner does not need to handle exploration on its own. As described in Appendix E, our implementation therefore does not enforce the learner to be exploratory, and allows the learner to remain deterministic. Concretely, we share the standard-deviation parameters between the guider and learner and stop gradients through this shared standard deviation when computing the KL-divergence term; only the means are updated by the KL loss.
> This design makes the learner **unsuitable as a behavioral policy**, because it cannot maintain meaningful stochastic exploration.
>
> For this reason, although exploring an annealing-based behavioral policy is promising, it would require a substantially different implementation design—particularly around exploration—and a comprehensive empirical evaluation, which we leave for future work.
>
> > [**Weakness 2**] It would be nice to have some more task-specific qualitative analysis / interesting behaviors / visual figures in the experimental section.
>
> We appreciate the suggestion. Unfortunately, for our tasks we did not identify qualitative visualizations that provide meaningful additional insights.
> That said, we have added **additional experimental figures (Fig. 9–12)** in the appendix. Notably, Fig. 12 visualizes how the guider–learner KL is constrained and how the temperature parameter α adapts to enforce this constraint, which may partially address the reviewer’s concern.
>
>
> > [**Weakness 3**] Missing some references. The seminal work is Vapnik's SVM+[0]. Missing some privileged model-based rl works as well. Would like to see Vapnik properly cited in main work, and MBRL works could be supplemental in Appendix A.
>
> Thank you for pointing this out.
> We have included the missing citations—Vapnik’s SVM+ and relevant privileged model-based RL works—in the revised manuscript (main text and Appendix A).

---

> > ### Comment · Reviewer_KRDz · 2025-11-27
> > **Thanks for the response, opinion remains positive.**
> >
> > Thanks for the discussion on the on-policy data distribution. I remain positive on this paper and think it should be accepted given the substantial contributions of their method, theory, and empirical study. It is also well written and presented.
> >
> > There seems to be some confusion in the other reviewer discussions, that the student cannot imitate the privileged guider. I would suggest the authors to make it clear in the writing that the privileged imitation gap is directly addressed by this method through through the KL constraint in the guider objective.

---

### Comment · Area_Chair_zphc · 2025-11-26
**[ICLR 2026] Author-Reviewer Discussion Phase**

Dear Reviewers,

The authors have posted their rebuttal addressing your concerns. Please kindly review their response, as well as the comments from the other reviewers, and discuss any issues you believe remain unresolved. If the author response does not change your evaluation, please at least provide an acknowledgement indicating that you have carefully reconsidered it.

Thank you again for your dedication and effort in reviewing this submission.

Let’s have a constructive discussion!

Best regards,

Your AC

---

### Meta-Review · Area_Chair_fvf2 · 2025-12-18

**Summary:**

Reviewers broadly agree the paper is clearly motivated and empirically strong, proposing a guided policy-iteration style method that couples a privileged guider with a partially observable learner via KL alignment and off-policy actor–critic updates.

However, the core concern raised by Reviewer sD5w regarding Lemma 3.4 still feels insufficiently addressed. The reviewer’s main critique is the fundamental mismatch between the definitions of the guider and the learner(https://openreview.net/forum?id=f9u34Xxh6p&noteId=vHV6tKkC2G). While the authors clarified the convergence target is POMDP-optimal and emphasized the KL constraint’s role, they did not directly resolve this issue—this feels like a missed opportunity to address the reviewer’s core point head-on. Personally, I find this lack of a direct response somewhat unjustified.

Also, given that three reviewers initially gave scores below the acceptance threshold (Reviewers JrGq, sD5w, 8jcT all scored 4, with Reviewer Rgrc raising their score to 6 after rebuttal).

These issues lead to non-trivial remaining disagreement and prevent the paper from reaching the robustness expected for acceptance.

**Reviewer Concerns:**

1. Theory-to-practice soundness remains disputed: a core concern persisted that the theoretical story around Lemma 3.4 relies on strong alignment/convergence conditions.
2. The high complexity is a barrier to reproduction and adoption.

**Reviewer Scores:**

Reviewer KRDz: 8.
Reviewer JrGq: 4.
Reviewer Rgrc:4->6.
Reviewer sD5w:4.
Reviewer 8jcT: 4.

---

### Decision · Program_Chairs · 2026-01-26

Reject